# SPRINT: SPARSE-DENSE RESIDUAL FUSION FOR EFFICIENT DIFFUSION TRANSFORMERS

**Dogyun Park**[1,2]* **Moayed Haji-Ali**[1] **Yanyu Li**[1] **Willi Menapace**[1]
**Sergey Tulyakov**[1] **Hyunwoo J. Kim**[3] **Aliaksandr Siarohin**[1] **Anil Kag**[1]
[1]Snap Inc. [2]Korea University [3]KAIST

Project page: https://snap-research.github.io/Sprint

## ABSTRACT

Diffusion Transformers (DiTs) deliver state-of-the-art generative performance but their quadratic training cost with sequence length makes large-scale pretraining prohibitively expensive. Token dropping can reduce training cost, yet naïve strategies degrade representations, and existing methods are either parameter-heavy or fail at high drop ratios. We present **SPRINT** (**Sp**arse–Dense **R**esidual Fus**i**on for Efficient Diffusion **T**ransformers), a simple method that enables aggressive token dropping (up to $75\%$) while preserving quality. SPRINT leverages the complementary roles of shallow and deep layers: early layers process all tokens to capture local detail, deeper layers operate on a sparse subset to cut computation, and their outputs are fused through residual connections. Training follows a two-stage schedule: long masked pre-training for efficiency followed by short full-token fine-tuning to close the train–inference gap. On ImageNet-1K $256^2$, SPRINT achieves $9.8\times$ training savings with comparable FID/FDD, and at inference, its **Path-Drop Guidance (PDG)** nearly halves FLOPs while improving quality. These results establish SPRINT as a simple, effective, and general solution for efficient DiT training.

## 1 INTRODUCTION

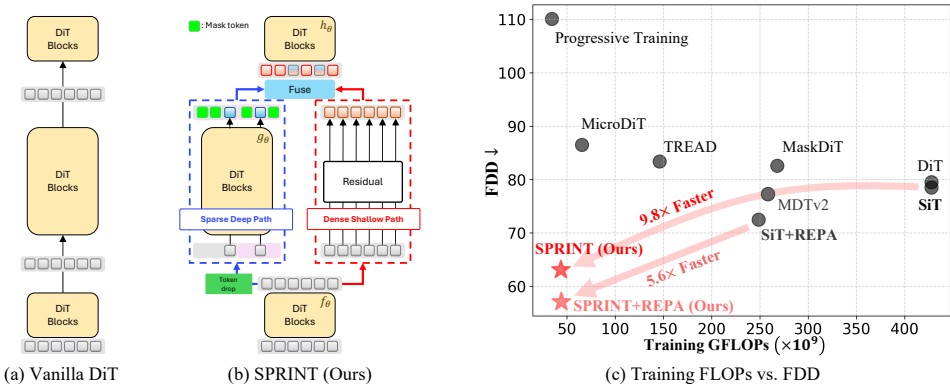

Figure 1: **Sparse–dense residual fusion improves the efficiency of diffusion transformer training.** SPRINT decouples the computationally heavy middle blocks of DiT into a sparse–deep path and a dense–shallow residual path. Notably, SPRINT achieves up to **5.6**$\times$ and **9.8**$\times$ lower training cost compared to vanilla models, *while improving generation quality.*

Diffusion Transformers (DiTs) (Peebles & Xie, 2023; Esser et al., 2024b) have emerged as a powerful class of generative models (OpenAI, 2024; Labs, 2024a). Yet their training cost scales quadratically with sequence length, making large-scale pretraining prohibitively expensive in compute and memory. A natural way to reduce training cost is to shorten sequences by dropping tokens during training. However, naïve token dropping (Sehwag et al., 2025) degrades representations and leads to poor generalization when models are evaluated with full-token inputs at inference.

---

*Work done during internship at Snap Inc.

Another direction is to guide DiTs with external supervision. For instance, REPA (Yu et al., 2024) aligns intermediate DiT features with DINOv2, accelerating convergence. However, such auxiliary losses can harm long-term performance or destabilize training (Wang et al., 2025), since pre-trained vision features are not naturally aligned with diffusion's iterative denoising. Recent work (Zheng et al., 2024; Gao et al., 2023) has explored more advanced token-dropping strategies. While promising, these methods either add substantial parameters (Sehwag et al., 2025) or only support moderate drop ratios (Krause et al., 2025; Zheng et al., 2024), and break down under aggressive settings (e.g., 75%).

In this work, we present a training algorithm that enables high-ratio token dropping while preserving robust, semantically meaningful representations that transfer effectively to full-token fine-tuning. Our design philosophy is to train DiTs efficiently with minimal architectural changes, achieving performance on par with—or better than—strong baselines. The core idea is to exploit the complementary roles of shallow and deep layers in neural networks: shallow layers capture fine-grained local details, while deeper layers model global semantics. However, in standard DiT training, deeper layers often waste computation on redundant local details that contribute little to modeling global semantics, due to the homogeneous architecture of DiTs. This redundancy significantly slows training convergence and reduces efficiency. We demonstrate that reformulating the architecture and coupling it with a principled token-dropping strategy resolves this issue.

**Our Solution.** We introduce *Sparse–Dense Residual Fusion for Efficient Diffusion Transformers* (**SPRINT**), a simple strategy that enables aggressive token dropping while preserving representation quality. Specifically, we partition the DiT into three components: encoder, middle blocks, and decoder. The encoder processes all tokens to encode local information, producing dense shallow features. Before the middle blocks, we drop most tokens (typically 75%), forcing deeper layers to focus on sparse global context with far lower compute, making sparse deep features. Simple residual fusion mechanism then combines dense shallow features with sparse deep features, while dummy masking tokens ensure dimensional alignment, and the fused representation is passed to the decoder.

Training proceeds in two stages. First, we perform long pre-training with 75% token dropping, yielding large compute savings. Then, a short fine-tuning stage restores full-token processing in the middle blocks, allowing them to adapt to dense inputs and closing the train–inference gap. Training uses the standard diffusion loss, and the DiT block design remains unchanged, making SPRINT easy to integrate into existing codebases.

Notably, the dual-path structure of SPRINT (dense shallow and sparse deep) enables a surprisingly efficient guidance sampling strategy, which we denote as *Path-Drop Guidance* (PDG). Standard classifier-free guidance requires two full forward passes of the model to compute conditional and unconditional estimates, thereby doubling inference cost. In contrast, under our framework, we can efficiently obtain the unconditional estimate by entirely bypassing the middle blocks and using only the dense shallow path. We demonstrate that PDG reduces the cost of guidance sampling by nearly 50% *while improving generation quality*.

**Contributions.** Our work makes the following key contributions:

- We propose *Sparse-Dense Residual Fusion* (SPRINT), which fuses dense shallow and sparse deep features for efficient DiT training, supporting up to 75% token dropping and yielding large efficiency gains over prior methods (Tab. 1, Fig. 3c).

- We demonstrate faster convergence and improved efficiency on modern DiTs. On ImageNet-1K $256^2$ class-conditional generation, SPRINT reduces training GFLOPs by **9.8×** compared to standard SiT training while achieving similar or better quality (Fig. 1c, Tab. 3).

- SPRINT provides new insights into DiT representations: our dense–shallow features are more noise-invariant and semantically expressive (Fig. 6); achieve higher CKNNA scores than vanilla DiT (Fig. 3b); and shallow versus deep paths specialize in local versus global semantics (Fig. 4).

- We introduce *Path-Drop Guidance* (PDG), a replacement for classifier-free guidance (CFG) that computes the unconditional pass using only dense shallow features. PDG nearly halves inference FLOPs while surpassing CFG in generation quality (Fig. 2, Tab. 3).

- We show that SPRINT is simple, architecture-agnostic, and complementary to alignment-based methods. It applies seamlessly across architectures (SiT, UViT), latent spaces (SD, FLUX VAE), and resolutions (256, 512), and provides further gains when combined with REPA (Yu et al., 2024).

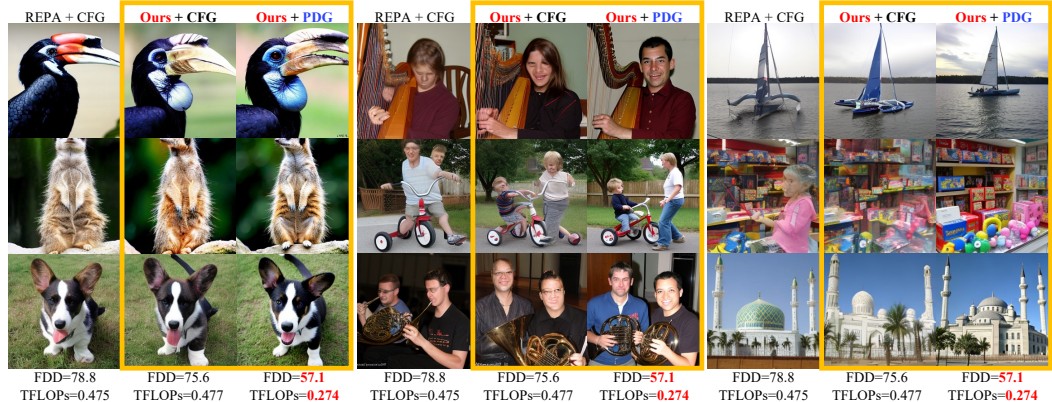

Figure 2: **SPRINT improves visual quality over baseline with only 57% of inference FLOPs**. We present samples from two SiT-XL/2$_{\text{REPA}}$ models after 1M training iterations, where SPRINT is applied to one of the models. For our approach, we further incorporate the proposed Path-Drop Guidance (PDG), yielding improved FDD scores and higher visual quality compared to vanilla REPA.

## 2 RELATED WORK

**Accelerating DiT training via representation alignment.** Several works accelerate DiT convergence by aligning internal features with pre-trained vision transformers. REPA (Yu et al., 2024) aligns intermediate DiT activations with DINOv2 features, while Lee et al. (2025) extend this to text–image models via a contrastive loss. Wang & He (2025) instead propose a dispersive loss that spreads features without external alignment. However, HASTE (Wang et al., 2025) shows that alignment signals can conflict with diffusion objectives and destabilize training. These objectives are complementary to our token-dropping scheme and can be combined to further boost performance (Tab. 2).

**Efficient DiT training with token dropping.** Another direction reduces training cost by shortening sequences. Progressive training (Podell et al., 2024; Esser et al., 2024b) first pre-trains at 128×128 before fine-tuning at 256×256. MDTv2 (Gao et al., 2023) restructures DiT into an encoder–decoder, processing masked tokens with skip connections and optimizing both reconstruction and diffusion losses. MaskDiT (Zheng et al., 2024) drops random patches, replaces them with mask tokens, and trains an auxiliary decoder, which adds inference cost. MicroDiT (Sehwag et al., 2025) adds a patch-mixer for high masking ratios; and TREAD (Krause et al., 2025) bypasses subsets of tokens through inner layers to optimize full denoising loss. These approaches work at moderate drop ratios ($\leq 50\%$) but degrade at aggressive settings (*e.g.*, 75%) and are difficult to pair with alignment losses. In contrast, our approach remains alignment-friendly and robust even under high drop rates.

## 3 SPRINT: SPARSE-DENSE RESIDUAL FUSION FOR EFFICIENT DIFFUSION TRANSFORMERS

### 3.1 PRELIMINARIES

**Diffusion and flow-based generative models.** Diffusion and flow-based models (Ho et al., 2020; Song et al., 2020; Lipman et al., 2023; Liu et al., 2023) learn a continuous transformation between a simple reference distribution $\pi_1$ (e.g., Gaussian noise) and a target data distribution $\pi_0$. Given $\mathbf{x}_0 \sim \pi_0$ and $\mathbf{x}_1 \sim \pi_1$, the transformation evolves over $t \in [0, 1]$ by the ODE

$$\frac{\mathrm{d}\mathbf{x}_t}{\mathrm{d}t} = v(\mathbf{x}_t, t), \tag{1}$$

where $\mathbf{x}_t$ interpolates between $\mathbf{x}_0$ and $\mathbf{x}_1$, and $v : \mathbb{R}^d \times [0, 1] \to \mathbb{R}^d$ is the velocity field. We use $\mathbf{x}_t \sim \mathcal{N}(\alpha_t \mathbf{x}_0, \sigma_t^2 I)$ with $\alpha_0 = \sigma_1 = 1$, $\alpha_1 = \sigma_0 = 0$, and adopt a linear schedule (Ma et al., 2024): $\alpha_t = 1 - t$, $\sigma_t = t$. A neural network $v_\theta$ (e.g., DiT) learns $v$ by minimizing

$$\min_\theta \ \mathbb{E}_{\mathbf{x}_0, \mathbf{x}_1, t}\big[\|v(\mathbf{x}_t, t) - v_\theta(\mathbf{x}_t, t)\|^2\big]. \tag{2}$$

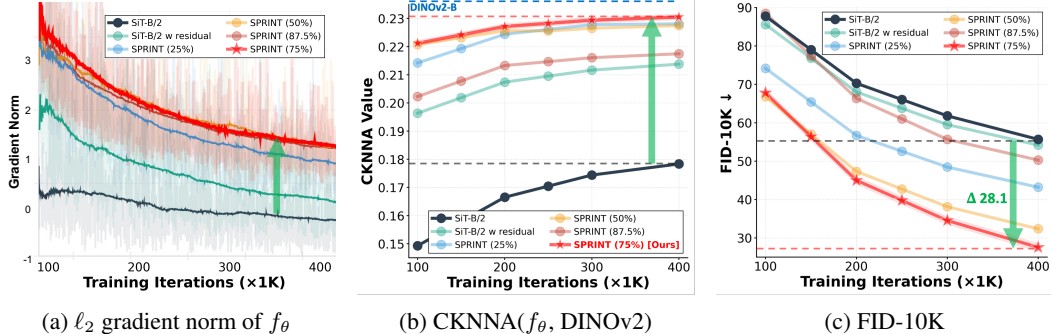

(a) $\ell_2$ gradient norm of $f_\theta$      (b) CKNNA($f_\theta$, DINOv2)      (c) FID-10K

Figure 3: **Training behavior of diffusion transformers.** We empirically analyze the training dynamics of SiT-B/2 and its SPRINT variants under different token-drop ratios. **(a)** We measure the $\ell_2$ gradient norm of $f_\theta$, showing that SPRINT enables the encoder to receive stronger gradient signals from the loss. **(b)** SPRINT variants achieve higher and earlier CKNNA scores than SiT, indicating SPRINT learns more semantic, noise-robust representations. **(c)** SPRINT converges substantially faster and to lower FID than SiT, with the gap further widening at higher drop ratios (up to 75%), highlighting both the effectiveness and efficiency of our framework.

**Token dropping in diffusion transformers.** Given a noisy image $\mathbf{x}_t$, a DiT divides it into non-overlapping $p \times p$ patches, producing tokens $\mathbf{x}_t \in \mathbb{R}^{B \times N \times D}$, where $N = \frac{HW}{p^2}$, $D$ is the embedding dimension, and $H \times W$ the image resolution. Since the attention cost in DiTs scales quadratically with $N$, dropping tokens reduces training cost. For a drop ratio $r$, we remove $\lfloor rN \rfloor$ tokens and process only the remaining $N - \lfloor rN \rfloor$ with DiT blocks. Although described for 2D images, this naturally extends to other modalities such as video.

## 3.2 BOTTLENECK IN STANDARD DiT TRAINING

Standard Diffusion Transformers (DiTs) use a homogeneous architecture where every layer, from shallow to deep, processes the full set of dense tokens. This is inefficient: in deeper layers, token representations become redundant as features shift from local, high-frequency patterns to global, low-frequency semantics (Hoover et al., 2019; Voita et al., 2019). Inference-time pruning and merging methods (Rao et al., 2021; Chang et al., 2023; Bolya & Hoffman, 2023) further show that large fractions of tokens can be removed in later layers with minimal effect on output quality. Training deep layers on all tokens thus wastes compute, spending a large portion of the FLOP budget on fine-grained details that contribute little to modeling global structure.

We address this by introducing architectural specialization: 1. **Early layers** process *dense* tokens to robustly capture local evidence under noisy input and build a rich foundation of features. 2. **Deeper layers** operate on a *sparse* subset of tokens to efficiently model global semantic relationships without redundant computation. 3. **Final layers** reintroduce all tokens for dense prediction. Based on these principles, we reformulate the DiT architecture with a dense–sparse fusion mechanism.

## 3.3 SPARSE–DENSE RESIDUAL FUSION

We propose *Sparse–Dense Residual Fusion for Efficient Diffusion Transformers* (SPRINT), which decouples dense local details from sparse global semantics, improving efficiency by accelerating convergence and reducing compute. An overview is shown in Fig. 1. We begin with a standard DiT, divided into encoder $f_\theta$ (first two blocks), middle blocks $g_\theta$, and decoder $h_\theta$ (final two blocks), and reformulate the computation flow as:

1. **Encoder** $f_\theta$ processes all noisy tokens to produce a feature map that retains fine-grained local noise information across all spatial locations.

2. **Dense shallow path** creates a residual connection that directly forwards the dense feature map from $f_\theta$ to the fusion block, preserving local, high-frequency detail.

3. **Sparse deep path** drops a large fraction of tokens (e.g., 75%) before $g_\theta$, forcing the deep layers to operate on a sparse subset, yielding sparse global context.

4. **Fusion and decoder** integrate dense local information from the shallow path with sparse global context from the deep path to predict all tokens.

Formally, given input tokens $\mathbf{x}_t \in \mathbb{R}^{B \times N \times C}$, we first compute dense features $\mathbf{f}_t = f_\theta(\mathbf{x}_t)$. A fraction $r$ of tokens (the *drop ratio*) is removed to form $\mathbf{f}_t^{\text{drop}}$, which is processed by the middle blocks: $\mathbf{g}_t^{\text{drop}} = g_\theta(\mathbf{f}_t^{\text{drop}})$. To fuse the dense and sparse paths, we restore $\mathbf{g}_t^{\text{drop}}$ to the original sequence length by padding the dropped positions with a fixed [MASK] token (denoted $\mathbf{M}$), yielding $\mathbf{g}_t^{\text{pad}} \in \mathbb{R}^{B \times N \times C}$. We concatenate $\mathbf{f}_t$ and $\mathbf{g}_t^{\text{pad}}$ along the channel dimension, project back to the original size, and feed the fused representation to the decoder $h_\theta$. This enables the decoder to combine local details from the encoder with sparse global semantics from the middle blocks for full-token prediction. The entire model is trained end-to-end by minimizing the flow matching loss in Eq. 2 (refer to Alg. 1).

**Improving training efficiency with minimal modification.** SPRINT improves training through two key mechanisms. First, it reduces *per-iteration compute cost* by restricting the expensive middle blocks $g_\theta$ to a sparse token set, while the dense shallow path preserves fine-grained information. Unlike prior methods, it remains stable even under aggressive drop ratios (Fig. 3c) where others fail. Second, it *accelerates iteration-wise convergence* by enhancing a contextual and relation learning: the decoder $h_\theta$ must predict all tokens despite most deep-path inputs being [MASK] tokens. This forces encoder ($f_\theta$) and middle blocks ($g_\theta$) to learn robust, context-aware features, as reflected in faster FID improvement (Fig. 3c), stronger gradient flow (Fig. 3a), and richer representations (CKNNA in Fig. 3b). These gains come with *minimal architectural change*: the standard DiT blocks remain intact, making SPRINT easy to integrate into existing codebases. Analysis details are in Appendix A.

**Dense–shallow vs. sparse–deep features.** The ablation in Fig. 4 highlights their complementary roles. The *dense–shallow path* preserves local textures (e.g., feathers, skin patterns) but fails to form coherent global structure. The *sparse–deep path* captures global shapes (e.g., bird outline, shark body) but introduces severe texture artifacts. Fusing both yields high-quality outputs with realistic global semantics and fine local detail, showing that dense–shallow features encode *local evidence* while sparse–deep features capture *global semantics*.

| Both path | w/o sparse path | w/o dense path | Both path | w/o sparse path | w/o dense path |

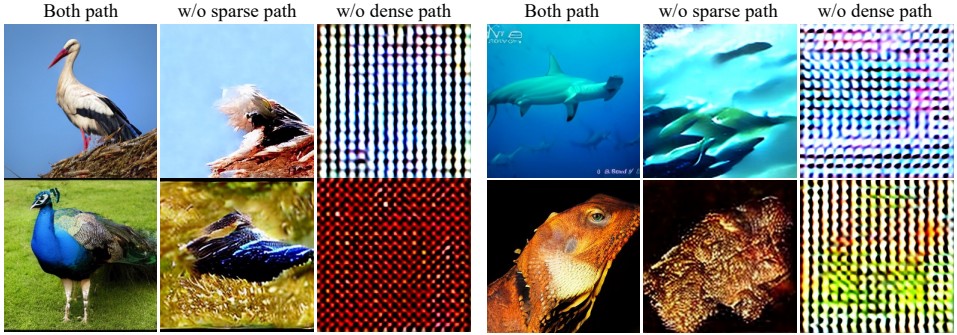

Figure 4: **Roles of dense–shallow and sparse–deep features.** Dense–shallow features preserve local textures but lose global structure, while sparse–deep features capture global shapes but distort local details. Fusing both recovers high-quality outputs with coherent semantics and fine detail.

**Fine-tuning with full tokens.** After efficient sparse pre-training, we transition the middle blocks to operate on the full token set for a brief fine-tuning stage, addressing the potential train–inference gap as demonstrated in prior works (Zhang et al., 2024; Sehwag et al., 2025; Krause et al., 2025) (refer to Alg. 2). Since pre-training typically dominates with 1M–4M iterations, this fine-tuning phase is short (e.g., 100K–200K iterations), yet sufficient for the deeper layers to adapt to the full data distribution, ensuring high inference quality while retaining most of the pre-training speedup.

### 3.4 EFFICIENT PATH-DROP GUIDANCE (PDG)

SPRINT's dual-path design also enables efficient guidance during inference. Standard Classifier-Free Guidance (CFG) doubles sampling cost by requiring two forward passes per step: one conditional $\mathbf{v}_\theta(\mathbf{x}_t, \mathbf{c})$ and one unconditional $\mathbf{v}_\theta(\mathbf{x}_t, \emptyset)$. Auto Guidance (Karras et al., 2024a) shows that the unconditional pass can be replaced by a weaker network. The SPRINT architecture inherently contains a natural weaker network: the dense shallow path that bypasses the deep middle blocks. We therefore introduce **Path-Drop Guidance** (PDG): For the conditional estimate, we perform a full forward pass. For the unconditional estimate, we bypass $g_\theta$ entirely, replacing it with [MASK]

Table 1: **Training efficiency on ImageNet 256$^2$**. Iteration-wise results of different token-dropping methods *with same 75% dropping rate*. We report total training TFLOPs (using Deepspeed library) and performance with/without classifier-free guidance, along with SPRINT's relative gains over SiT (**Gain** $\Delta$). All methods use 50 sampling steps with ODE sampler.

| Method | AE | TFLOPs ($\times 10^6$) | w/o CFG ($w = 1.0$) | | | | | w CFG ($w = 1.4$) | | | | |
|---|---|---|---|---|---|---|---|---|---|---|---|---|
| | | | FDD ↓ | FID ↓ | IS ↑ | Pre. ↑ | Rec. ↑ | FDD ↓ | FID ↓ | IS ↑ | Pre. ↑ | Rec. ↑ |
| *400K training iterations* | | | | | | | | | | | | |
| Improved SiT-XL/2 | SD | 24.4 | 351.1 | 12.8 | 97.4 | 0.66 | **0.65** | 185.0 | 3.09 | 211.6 | 0.81 | 0.55 |
| + Progressive Training | SD | **16.8** | 365.5 | 12.7 | 96.2 | 0.67 | 0.63 | 215.6 | 3.47 | 206.4 | **0.83** | 0.53 |
| + MDTv2 | SD | 21.2 | 558.5 | 21.1 | 68.9 | 0.61 | 0.63 | 366.5 | 5.61 | 176.3 | 0.76 | 0.54 |
| + MicroDiT | SD | 20.8 | 349.9 | 11.5 | 99.9 | 0.67 | 0.64 | 178.1 | 3.16 | 213.7 | 0.82 | 0.54 |
| + Tread | SD | 19.7 | 461.1 | 16.3 | 89.9 | 0.63 | 0.64 | 264.3 | 4.07 | 201.2 | 0.80 | 0.54 |
| + **SPRINT** (Ours) | SD | 18.7 | **262.6** | **9.30** | **118.5** | **0.68** | **0.65** | **136.5** | **2.56** | **247.1** | 0.82 | **0.56** |
| **Gain** $\Delta$ | | ×**1.32** | **+88.5** | **+3.5** | **+24.1** | | | **+48.5** | **+0.53** | **+35.5** | | |
| *1M training iterations* | | | | | | | | | | | | |
| Improved SiT-XL/2 | SD | 61.2 | 290.0 | 10.9 | 113.4 | 0.66 | **0.67** | 146.0 | 2.36 | 243.7 | 0.80 | 0.58 |
| + Progressive Training | SD | **25.8** | 359.4 | 12.3 | 102.2 | 0.67 | 0.65 | 188.1 | 2.95 | 222.2 | **0.82** | 0.55 |
| + MDTv2 | SD | 39.2 | 522.7 | 18.8 | 77.2 | 0.61 | 0.64 | 326.7 | 4.68 | 183.1 | 0.77 | 0.55 |
| + MicroDiT | SD | 37.5 | 293.4 | 10.9 | 113.8 | **0.68** | 0.65 | 147.6 | 2.53 | 241.4 | **0.82** | 0.55 |
| + Tread | SD | 34.5 | 372.6 | 12.3 | 112.1 | 0.66 | 0.66 | 197.7 | 2.82 | 242.9 | 0.80 | 0.57 |
| + **SPRINT** (Ours) | SD | 31.5 | **248.8** | **9.15** | **129.5** | 0.67 | **0.67** | **126.1** | **2.29** | **268.3** | 0.81 | **0.59** |
| **Gain** $\Delta$ | | ×**1.94** | **+41.2** | **+1.75** | **+16.1** | | | **+14.9** | **+0.07** | 24.6 | | |
| *400K training iterations* | | | | | | | | | | | | |
| Improved SiT-XL/2 | Flux | 24.6 | 358.9 | 14.8 | 84.4 | 0.64 | 0.63 | 178.7 | 3.95 | 210.7 | 0.83 | 0.50 |
| + Progressive Training | Flux | **17.0** | 375.3 | 13.5 | 89.2 | **0.66** | 0.63 | 186.3 | 4.02 | 205.4 | **0.84** | 0.49 |
| + MicroDiT | Flux | 20.9 | 420.9 | 17.8 | 76.8 | 0.61 | **0.64** | 212.9 | 4.45 | 196.2 | 0.81 | **0.51** |
| + Tread | Flux | 19.8 | 470.1 | 19.9 | 72.2 | 0.60 | 0.63 | 255.0 | 5.18 | 187.5 | 0.79 | 0.50 |
| + **SPRINT** (Ours) | Flux | 18.8 | **268.4** | **11.4** | **101.9** | **0.66** | 0.63 | **135.4** | **3.77** | **239.8** | 0.83 | **0.51** |
| **Gain** $\Delta$ | | ×**1.31** | **+90.2** | **+3.4** | **+17.5** | | | **+43.3** | **+0.18** | **+29.1** | | |

tokens. Formally, the conditional and unconditional velocities are:

$$v(\mathbf{x}_t, \mathbf{c}) = h_\theta(\text{Fusion}(g_\theta(f_\theta(\mathbf{x}_t, \mathbf{c})), f_\theta(\mathbf{x}_t, \mathbf{c})), \mathbf{c}), \quad (3)$$

$$v(\mathbf{x}_t, \emptyset) = h_\theta(\text{Fusion}(\mathbf{M}, f_\theta(\mathbf{x}_t, \emptyset)), \emptyset), \quad (4)$$

where $\mathbf{M}$ denotes the [MASK] token tensor. This provides high-quality generation while nearly halving FLOPs and latency per step, since the expensive middle blocks are executed only once.

### 3.5 STRUCTURED GROUP-WISE TOKEN SUBSAMPLING

The effectiveness of token dropping depends not just on how many tokens are removed, but on which are kept. Uniform random sampling risks leaving large contiguous holes in the feature map. To avoid this, we propose a *structured group-wise subsampling* strategy that guarantees local coverage while maintaining global irregularity. Specifically, we partition tokens into small, non-overlapping groups in their native topology (*e.g.*, 2D for images). For images, we divide the $(H/p) \times (W/p)$ grid into $n \times n$ groups. At each training iteration, we randomly select $k$ tokens per group, giving a drop ratio $r = 1 - k/n^2$. We use $n = 2$, $k = 1$, corresponding to a 75% drop ratio. This ensures that every local patch is represented while preventing the model from overfitting to fixed sampling patterns.

## 4 EXPERIMENT

### 4.1 EXPERIMENTAL DETAILS

**Training details.** Our framework follows the setups of DiT (Peebles & Xie, 2023) and SiT (Ma et al., 2024). Unless stated otherwise, most of the experiments are trained on ImageNet-1K at $256 \times 256$ resolution using pretrained VAEs from Stable Diffusion (Rombach et al., 2022) and Flux (Labs, 2024b), both with $8\times$ downsampling but encoding into 4 and 16 channels, respectively. Unless stated otherwise, models are pre-trained with a 75% token drop ratio using our structured group-wise subsampling. We adopt the SiT architecture, where each block contains a self-attention and a feed-forward layer, and apply standard improvements: RMS Normalization for queries and keys (Touvron et al., 2023a;b), 2D RoPE for positional embeddings (Wang et al., 2024), and lognormal timestep sampling (Esser et al., 2024a). Experiments focus on SiT-B/2 and SiT-XL/2. Additional hyperparameters and training details are provided in Appendix C. Pre-training and fine-tuning algorithm is provided in Alg. 1 and 2, respectively.

Table 2: **Compatibility with other architectures.** We apply SPRINT to REPA and U-ViT on the SD autoencoder, reporting performance at 400K iterations with/without classifier-free guidance and SPRINT's relative gains over SiT (**Gain** $\Delta$). All metrics use 50 ODE sampling steps.

| Method | w/o CFG ($w = 1.0$) | | | | | w CFG ($w = 1.4$) | | | | |
|---|---|---|---|---|---|---|---|---|---|---|
| | FDD ↓ | FID ↓ | IS ↑ | Pre. ↑ | Rec. ↑ | FDD ↓ | FID ↓ | IS ↑ | Pre. ↑ | Rec. ↑ |
| Improved SiT-XL/2$_{REPA}$ | 279.6 | 10.0 | 114.0 | 0.67 | 0.66 | 146.6 | 2.42 | 237.1 | **0.81** | 0.57 |
| + **SPRINT** (Ours) | **234.5** | **8.68** | **129.6** | 0.67 | **0.67** | **125.1** | **2.38** | **259.8** | 0.80 | **0.59** |
| **Gain** $\Delta$ | **+45.1** | **+1.32** | **+15.6** | | | **+21.5** | **+0.04** | **+22.7** | | |
| Improved U-ViT-XL/2 | 335.1 | 12.1 | 98.6 | 0.67 | 0.64 | 193.7 | 3.36 | 200.3 | 0.80 | **0.56** |
| + **SPRINT** (Ours) | **271.7** | **9.20** | **114.4** | **0.69** | 0.64 | **146.4** | **2.97** | **236.7** | **0.83** | 0.54 |
| **Gain** $\Delta$ | **+63.4** | **+2.9** | **+15.8** | | | **+30.1** | **+0.39** | **+36.4** | | |

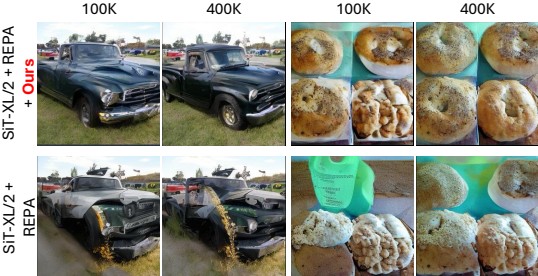

Figure 5: **SPRINT improves visual scaling.** Qualitative comparison of images generated without classifier-free guidance at 400K iterations using two SiT-XL/2$_{REPA}$ models, with SPRINT applied to the model in the upper row.

Figure 6: **SPRINT improves feature semantics.** We visualize the PCA features of $f_\theta$ and $g_\theta$ from two SiT-XL/2 models at 400K iterations, with SPRINT applied to the model in the upper row.

**Evaluation details.** We evaluate generation quality using standard metrics: FDD (Fréchet Distance on DINOv2 (Oquab et al., 2023) features), FID (Fréchet Inception Distance (Heusel et al., 2017)), Inception Score (IS) (Salimans et al., 2016), Precision, and Recall (Kynkäänniemi et al., 2019). Among these, FDD has been shown to be more reliable for diffusion models (Stein et al., 2023). To assess training and inference efficiency, we report total training FLOPs and inference FLOPs computed with the DeepSpeed library. We provide details in Appendix C, D. Inference algorithm is provided in Alg. 3.

## 4.2 SYSTEM-LEVEL COMPARISON

We compare against following methods to demonstrate the effectiveness and efficiency of our method:

1. **Dense SOTA:** We use SiT (Ma et al., 2024) models as our primary baseline. This represents a state-of-the-art model trained with full, dense tokens, providing a direct measure of the trade-off between performance and our efficiency gains.

2. **Sparse SOTA:** We compare against recent methods that leverage a token-dropping strategy to accelerate training, e.g., MicroDiT (Sehwag et al., 2025) and Tread (Krause et al., 2025).

3. **Alternative methods:** We also compare against progressive training, a popular strategy where a model is pretrained on a lower resolution and finetuned on the full resolution.

A more detailed description of each baseline is provided in Appendix E. We adopt the same architectural improvements for all models and report the performance during training in Tab. 1. SPRINT demonstrates superior performance and efficiency across all settings. At just 400K training iterations with the SD-VAE, our model significantly outperforms SiT-XL/2 baseline (e.g., a +88.5 improvement in FDD) while using 1.32× fewer FLOPs. As training progresses to 1M iterations, SPRINT consistently improves over the baseline while becoming even more efficient, achieving a 1.95× computational speedup. This result highlights SPRINT's dual acceleration: achieving higher sample quality in the same training iterations with lower computational budget. In contrast, competing token-dropping methods fail to match the performance of SiT-XL/2, even at a *higher* computational cost than our method. These trends hold when using classifier-free guidance.

Table 3: **Comprehensive comparison on ImageNet** $256 \times 256$ class-conditioned generation with classifier-free guidance. ↓ / ↑ indicate whether lower or higher values are better, respectively. * denotes training with batch size 1024, † *our reproduction* with architectural improvements, and ‡ use of guidance scheduling. Metrics are evaluated with 250 sampling steps using the SDE sampler. TFLOPs are measured with the DeepSpeed library (refer to Appendix D for details.)

| Method | Epochs | #Params. | Training TFLOPs ↓ ($\times 10^6$) | Inference TFLOPs ↓ | FDD ↓ | FID ↓ | Pre. ↑ | Rec. ↑ |
|---|---|---|---|---|---|---|---|---|
| ADM (Dhariwal & Nichol, 2021) | 400 | 673M | – | – | – | 3.94 | 0.82 | 0.52 |
| CDM (Ho et al., 2022) | 2160 | – | – | – | – | 4.88 | – | – |
| LDM-4 (Rombach et al., 2022) | 200 | 400M | – | – | – | 3.60 | 0.87 | 0.48 |
| U-ViT-H* (Bao et al., 2023) | 240 | 501M | – | – | – | 2.29 | 0.82 | 0.57 |
| DiT-XL (Peebles & Xie, 2023) | 1400 | 675M | 427.7 | 0.475 | 79.5 | 2.27 | 0.83 | 0.57 |
| FiTv2-XL (Wang et al., 2024) | 400 | 671M | – | – | 80.5 | 2.26 | 0.81 | 0.59 |
| MDTv2-XL (Gao et al., 2023) | 1080 | 742M | 258.3 | 0.521 | 77.3 | 1.86 | 0.81 | 0.60 |
| MDTv2-XL‡ (Gao et al., 2023) | 1080 | 742M | 258.3 | 0.521 | 75.2 | 1.58 | 0.79 | 0.65 |
| MaskDiT (Zheng et al., 2024) | 1600 | 730M | 268.0 | 0.513 | 82.4 | 2.28 | 0.80 | 0.61 |
| Tread (Krause et al., 2025) | 740 | 675M | 146.0 | 0.475 | – | 2.09 | 0.81 | 0.62 |
| SiT-XL (Ma et al., 2024) | 1400 | 675M | 427.7 | 0.475 | 78.5 | 2.06 | 0.82 | 0.59 |
| SiT-XL† | 400 | 675M | 122.2 | 0.474 | 79.5 | 2.04 | 0.82 | 0.60 |
| + SPRINT | 200 | 677M | **43.7** | 0.477 | 79.0 | 2.01 | 0.82 | 0.60 |
| + SPRINT | 400 | 677M | 65.1 | 0.477 | 75.4 | 1.96 | 0.80 | 0.61 |
| + SPRINT$_{PDG}$ | 400 | 677M | 65.1 | 0.274 | 58.4 | 1.62 | 0.80 | 0.63 |
| + SPRINT$_{PDG}^{‡}$ | 400 | 677M | 65.1 | **0.263** | **54.9** | **1.55** | 0.80 | 0.64 |
| SiT-XL$_{REPA}$ (Yu et al., 2024) | 800 | 675M | 248.6 | 0.475 | 72.5 | 1.80 | 0.81 | 0.61 |
| SiT-XL$_{REPA}^{†}$ | 200 | 675M | 62.1 | 0.474 | 78.8 | 1.93 | 0.81 | 0.60 |
| + SPRINT | 200 | 677M | **44.3** | 0.477 | 75.6 | 1.87 | 0.81 | 0.61 |
| + SPRINT$_{PDG}$ | 200 | 677M | 44.3 | 0.274 | 57.1 | 1.61 | 0.80 | 0.64 |
| + SPRINT$_{PDG}$ | 400 | 677M | 66.7 | 0.274 | 54.7 | 1.59 | 0.80 | 0.64 |
| + SPRINT$_{PDG}^{‡}$ | 400 | 677M | 66.7 | **0.263** | **49.6** | **1.49** | 0.81 | 0.64 |

**Generalization to other diffusion architectures.** To demonstrate that our method is a general training strategy and not limited to a specific DiT architecture, we apply SPRINT to two other prominent models: REPA (Yu et al., 2024) and U-ViT (Bao et al., 2023). We integrate our dense-sparse fusion mechanism into their respective backbones and report the results after 400K training iterations in Tab. 2. The results show that SPRINT provides significant improvements in all cases. When applied to REPA, SPRINT improves the FDD by +45.1 and FID by +1.32 (w/o CFG). Similarly, for U-ViT, we observe a +63.4 improvement in FDD and a +2.9 improvement in FID. These experiments confirm that SPRINT is a broadly applicable and effective method for accelerating the training.

**Visual analysis.** In Fig. 5, we show that SPRINT not only accelerates convergence quantitatively but also enhances the visual progression. At just 100K iterations, SPRINT produces coherent global structures (e.g., the shape of a car) along with fine details, whereas REPA lags behind. Furthermore, in Fig. 6, we analyze the PCA of features from $f_\theta$ and $g_\theta$, demonstrating that SPRINT learns more noise-invariant and semantically vivid representations than the SiT model across diffusion timesteps.

### 4.3 COMPARISON WITH STATE-OF-THE-ART MODELS

Tab. 3 compares SPRINT against recent state-of-the-art diffusion transformers. Our improved SiT closely matches the original SiT performance after 400 epochs (78.5 vs. 79.5 FDD). In contrast, SiT trained with SPRINT achieves comparable performance 79.0 FDD in only 200 epochs. At 400 epochs, SPRINT outperforms the improved SiT baseline by **4.4 FDD** (from 79.5 to 75.4) and **0.08 FID** while using just **53%** of the training FLOPs. This shows that SPRINT both accelerates convergence and substantially reduces training cost. At inference, Path-Drop Guidance (PDG) further boosts efficiency: with only 57% of the inference cost, SPRINT improves performance by **21.1 FDD** (from 79.5 to 58.4) over the improved SiT.

Similar trends hold when combined with REPA. SPRINT reduces FDD from 78.8 to 75.6 using only 71% of the training FLOPs. With PDG sampling at 400 epochs, it surpasses the official REPA model trained for 800 epochs by **17.8 FDD** and **0.21 FID**, while using only **27%** of the training FLOPs. Overall, SPRINT consistently improves generation quality while drastically lowering both training and inference cost, outperforming strong baselines and alignment-augmented models. Moreover, incorporating the recent guidance schedule (Kynkäänniemi et al., 2024) further boosts performance.

Table 4: Effect of token-drop strategies on FID.

| Strategy | FID ↓ |
|---|---|
| Random | 30.1 |
| Structured (Ours) | **27.5** |

Table 5: Effect of dense–sparse residuals on FID.

| Dense | Sparse | FID ↓ |
|---|---|---|
| ✗ | ✓ | 85.1 |
| ✓ | ✗ | 81.4 |
| ✓ | ✓ | **27.5** |

Table 6: Effect of $f_\theta, g_\theta, h_\theta$ on compute and performance.

| $f_\theta$ | $g_\theta$ | $h_\theta$ | FLOPs / iter ↓ | FID ↓ |
|---|---|---|---|---|
| 2 | 8 | 2 | **7.47G** | **27.5** |
| 3 | 6 | 3 | 9.33G | 29.1 |
| 5 | 2 | 5 | 13.1G | 49.2 |

Table 7: Effect of dense residuals and drop ratio $r$ on FID.

| Method | Dense residual | $r$ | FID ↓ |
|---|---|---|---|
| SiT-B/2 | ✗ | 0 | 55.6 |
| SPRINT | ✓ | 0 | 54.1 |
| | ✓ | 25% | 43.2 |
| | ✓ | 50% | 32.3 |
| | ✓ | **75%** | **27.5** |
| | ✓ | 87.5% | 50.2 |

Table 8: Effect of $f_\theta$ and $h_\theta$ depth on FID.

| $f_\theta$ | $g_\theta$ | $h_\theta$ | FID ↓ |
|---|---|---|---|
| 0 | 8 | 4 | 79.7 |
| 1 | 8 | 3 | 61.5 |
| 2 | 8 | 2 | **27.5** |
| 3 | 8 | 1 | 44.4 |
| 4 | 8 | 0 | 81.5 |

Figure 7: Effect of guidance scale on SiT and our SPRINT.

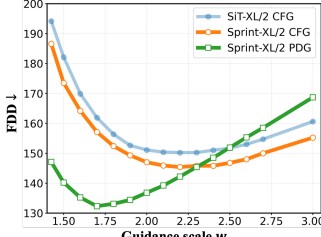

## 4.4 ANALYSIS

We mostly use SiT-B/2 configuration at 400K training iterations (detailed in Tab. 9) in following analysis unless stated otherwise.

**Sparse–dense residual fusion** (Tab. 5). To evaluate the importance of each path in sparse–dense residual fusion, we perform an ablation by disabling each of the two parallel paths during training. Removing the dense shallow path causes a sharp performance drop, with FID rising from 27.5 to 85.1, underscoring its role in accurate velocity prediction. Conversely, removing the sparse deep path reduces the model to a standard dense DiT with only four layers, which also degrades performance due to limited capacity. These results confirm that the parallel sparse–dense design is critical for maintaining high performance under token dropping.

**Token sampling strategy** (Tab. 4). We compare our structured group-wise sampling strategy with standard uniform random sampling. At the same 75% drop ratio, structured sampling improves FID from 30.1 to 27.5, demonstrating that preserving local coverage is crucial for effective sparse training.

**Effect of $g_\theta$ depth** (Tab. 6). We study the trade-off between performance and computation as a function of middle block depth. The default configuration yields the best FID (27.3) with the lowest cost (7.47G). Shifting layers from the middle block to the encoder and decoder (e.g., 3-6-3 or 5-2-5) increases cost without benefit, and FID degrades to 29.1 and 49.2, respectively. Thus, the default configuration strikes the best balance between efficiency and performance.

**Effect of $f_\theta$ and $h_\theta$ depth** (Tab. 8). We find that allocating at least two blocks to both $f_\theta$ (dense shallow path) and $h_\theta$ (sparse deep path) is critical for high performance. Reducing either to a single block already degrades results (FID 61.5 and 44.4). Moreover, entirely removing either block collapses performance (FID > 79): this supports our encoder (dense)–middle (sparse)–decoder (dense) design. The encoder must first operate on dense tokens to transform noisy inputs into noise-invariant features, after which the middle blocks can safely work on sparse tokens, and the decoder is applied after residual fusion. This is necessary for accurate prediction under high drop-ratio training.

**Drop ratio $r$** (Tab. 7). As the drop ratio increases from 0 to 75%, model performance steadily improves, with FID decreasing from 54.1 to 27.5. This trend indicates that higher sparsity in SPRINT promotes complementary interactions between the encoder and middle blocks, leading to more robust and efficient representations. However, at an extreme drop ratio of 87.5%, FID rises to 50.2, suggesting that excessive sparsity limites the model's representational capacity.

**Path-drop guidance** (Fig. 7). We compare FDD across guidance scales $w$ for CFG (SiT-XL/2), CFG (SPRINT), and PDG (SPRINT). PDG consistently outperforms both CFG baselines, achieving a lower (better) peak FDD. Moreover, it delivers these gains at nearly half the inference cost, since the unconditional estimate bypasses the middle blocks. These results show that PDG provides a superior trade-off, generating higher-quality samples while substantially reducing computational cost.

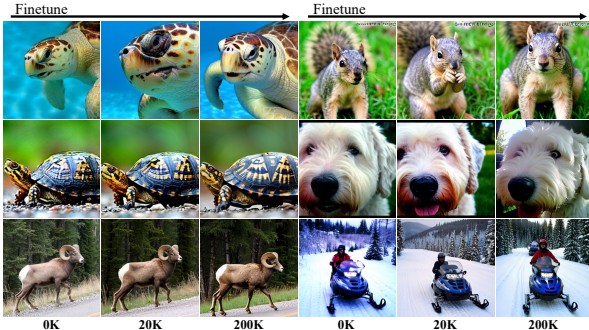
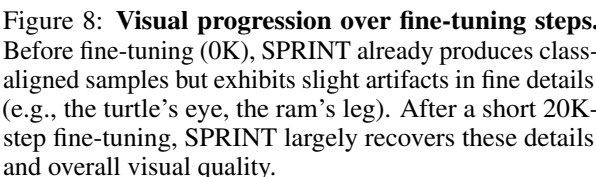
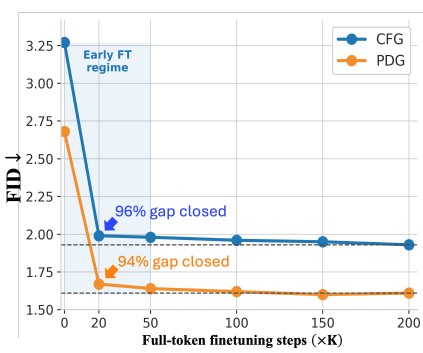

Figure 8: **Visual progression over fine-tuning steps.** Before fine-tuning (0K), SPRINT already produces class-aligned samples but exhibits slight artifacts in fine details (e.g., the turtle's eye, the ram's leg). After a short 20K-step fine-tuning, SPRINT largely recovers these details and overall visual quality.

Figure 9: **FID over fine-tuning steps** for CFG and PDG sampling. Just 20K fine-tuning steps recover over 94% of the 200K performance, indicating that a relatively short fine-tuning stage is sufficient to close the train–inference gap.

**Training at higher resolution** (Appendix F.1, Fig. 33). We also evaluate our model against baselines at $512^2$ resolution with XL config. Results are provided in Tab. 10 and show that SPRINT achieves 1.96 FID compared to 2.63 of SiT baseline with only 50% of training compute (184.8 vs. 366.6).

**Lower sampling steps** (Appendix F.2). SPRINT remains competitive at few-step inference, consistently surpassing SiT-XL/2 in Tab. 11. At 10 steps, it reduces FID from 7.37 to 6.29 and FDD from 205.2 to 174.5, highlighting the representational strength of our method.

## 4.5 BENEFITS OF FINE-TUNING

Here, we analyze the train–inference gap of SPRINT after the pre-training stage and the effect of the subsequent fine-tuning. Specifically, we perform qualitative and quantitative ablations over the number of fine-tuning steps after 2M pre-training iterations, reported in Fig. 8 and Fig. 9, respectively.

In Fig. 8, we observe that, before fine-tuning, SPRINT already produces class-aligned samples with globally coherent structure, but tends to miss some high-frequency details (e.g., the turtle's eye in second row, the ram's leg in third row), which is expected given that most tokens are dropped during pre-training. The role of the fine-tuning stage is therefore to recover these local details. Notably, after only 20K fine-tuning steps, SPRINT largely restores these details and improves overall visual quality. This observation is consistent with the quantitative trends in Fig. 9. For both CFG and PDG sampling, FID improvements beyond 50K fine-tuning iterations are marginal and eventually plateau. In particular, after just 20K steps, SPRINT recovers over 94% of the FID improvement achieved at 200K fine-tuning steps. This indicates that the majority of the train–inference gap closes very early—within 20K–50K iterations, corresponding to only 2.5% of the pre-training steps. This further confirms that SPRINT learns the necessary representations for high-quality generation during sparse pre-training, and that these representations transfer effectively to the full-token regime.

Overall, these results show that SPRINT is not overly sensitive to the precise length of the fine-tuning stage: a relatively short full-token fine-tuning is sufficient to recover the high-frequency details missing from sparse pre-training.

## 5 CONCLUSION

We introduced **SPRINT**, a simple and architecture-agnostic training framework for DiTs that combines dense–shallow and sparse–deep features through residual fusion. By exploiting the complementary strengths of shallow and deep layers, it enables aggressive token dropping (up to 75%) while preserving representation quality, and a two-stage schedule with masked pre-training and short full-token fine-tuning closes the train–inference gap. Experiments on ImageNet-1K show that SPRINT reduces training cost by up to $9.8\times$ while matching or surpassing the quality of strong baselines. SPRINT also enables **Path-Drop Guidance**, a simple replacement for CFG that halves inference cost while improving sample quality. Thus, SPRINT is a simple, effective, and general approach for efficient DiT training, applicable across architectures, resolutions, and alignment methods.

## REPRODUCIBILITY

We have made every effort to ensure the reproducibility of our results. Detailed hyper-parameters, training schedules, and architectural configurations are provided in the Appendix, including model definitions, pre-training fine-tuning iterations, number of sampling steps at inference, and compute resources. Our framework follows the well-established setups of DiT (Peebles & Xie, 2023) and SiT (Ma et al., 2024), which are widely adopted in diffusion research. Although our training code cannot be released at submission time, the use of these standardized setups, along with the provided experimental details, should allow independent reproduction of our results.

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

## A ANALYSIS DETAILS

**Training behavior (Fig. 3).** We provide the implementation details used to measure the training behavior shown in Fig. 3. We adopt the SiT-B/2 configuration from the SiT paper (Ma et al., 2024), which consists of 2 encoder blocks, 8 middle blocks, and 2 decoder blocks. In Fig. 3a, we plot the $\ell_2$ gradient norm of the encoder $f_\theta$ with respect to the flow-matching loss $\mathcal{L}$, i.e., $\|\nabla_{f_\theta}\mathcal{L}\|$, across pretraining iterations. This analysis highlights the improved gradient flow within the encoder blocks. Compared to the SiT baseline, SPRINT exhibits consistently stronger gradient propagation to the encoder as sparsity increases, leading to more effective parameter updates and faster convergence—reflected in both higher CKNNA scores and lower FID values.

In Fig. 3b, we report the Centered Kernel Nearest-Neighbor Alignment (CKNNA) (Huh et al., 2024) score, a relaxed variant of Centered Kernel Alignment (CKA). CKNNA is commonly used to assess the semantic alignment (Yu et al., 2024) between diffusion models and large-scale self-supervised visual encoders such as DINOv2. Intuitively, given a noisy input $\mathbf{x}_t$, CKNNA quantifies how well the intermediate features of a diffusion model capture noise-invariant semantics by comparing them with DINOv2 features extracted from the corresponding clean image $\mathbf{x}_0$. Higher CKNNA scores indicate more semantically meaningful and noise-robust representations that align more closely with the features of the visual encoder. We follow the definition and implementation provided in the original work (Huh et al., 2024). Specifically, we compute the CKNNA score between the output of the encoder $f_\theta$ on noisy inputs $\mathbf{x}_t$ and the output of DINOv2 on clean inputs $\mathbf{x}_0$. We randomly sample 10K images from the ImageNet-1K validation set and report results with $k = 10$.

Finally, in Fig. 3c, we report FID values computed with 10K generated images. Consistent with previous findings (Yu et al., 2024), we observe a strong negative correlation between the CKNNA values of intermediate diffusion features and FID scores. This suggests that higher alignment between diffusion features and high-quality visual representations leads to better generation quality.

**Roles of dense-shallow and sparse-deep features (Fig. 4).** In Fig. 4, we analyze the contribution of each path in SPRINT. To generate samples using only a single path, we replace the feature representation of one path with that of the other. In other words, we duplicate the features from one path and concatenate the original and duplicated features before feeding them into the decoder.

**PCA visualization of diffusion features (Fig. 6).** In Fig. 6, we perform a principal component analysis (PCA) of the intermediate features to better understand what the model has learned. PCA identifies the principal axes that capture the greatest variance in the feature space and is widely used to analyze representations learned by neural networks (Kim et al., 2022; Oquab et al., 2023; Park et al., 2024a; Ko et al., 2024). We compute PCA across patch embeddings and visualize the first three principal components as RGB channels. Specifically, we examine the outputs of the encoder $f_\theta$ and the middle blocks $g_\theta$ at different timesteps to observe how the feature representations evolve throughout the diffusion process. Additional PCA visualizations are provided in Fig. 34.

## B SPRINT WITH DIFFERENT DIFFUSION TRANSFORMERS

We provide details of the different diffusion transformers used in the main paper and describe how SPRINT is implemented on top of them.

**SiT (Ma et al., 2024).** We closely follow the architecture of SiT. The SiT model is structurally analogous to a Vision Transformer (ViT) (Dosovitskiy et al., 2020), consisting of a sequence of identical transformer blocks that process a patchified 1D token sequence. SiT adapts this for the diffusion task by incorporating timestep and class conditioning, which is injected into each block via AdaIN-zero layers. Because the architecture is a simple, homogeneous stack of blocks, it is straightforward to decouple it into our encoder, middle, and decoder blocks when applying SPRINT.

**REPA (Yu et al., 2024).** Representation Alignment (REPA) regularizes a DiT by aligning hidden states with clean image features from a pre-trained DINOv2 model. The architecture largely follows SiT, with the key modification being a projection layer inserted at the 8th transformer block to perform the alignment. To integrate SPRINT with REPA, we place this projection layer at the corresponding

Table 9: **Hyperparameters used for SPRINT.**

| | SiT-B+**SPRINT** (Fig. 3, Tab. 5-8) | SiT-XL+**SPRINT** (Tab. 1, 3) | SiT-XL$_{REPA}$+**SPRINT** (Tab. 2, 3) | SiT-XL+**SPRINT** (Tab. 10) | U-ViT-XL+**SPRINT** (Tab. 2) |
|---|---|---|---|---|---|
| *Architecture* | | | | | |
| Target latent res. | $32 \times 32$ | $32 \times 32$ | $32 \times 32$ | $64 \times 64$ | $32 \times 32$ |
| Patch size | 2 | 2 | 2 | 2 | 2 |
| Total Num. Layers | 12 | 28 | 28 | 28 | 28 |
| Num. $f_\theta$ Layers | 2 | 2 | 2 | 2 | 2 |
| Num. $g_\theta$ Layers | 8 | 24 | 24 | 24 | 24 |
| Num. $h_\theta$ Layers | 2 | 2 | 2 | 2 | 2 |
| Hidden dims | 384 | 1152 | 1152 | 1152 | 1152 |
| Num. heads | 6 | 16 | 16 | 16 | 16 |
| *Pretraining config.* | | | | | |
| Optimizer | AdamW | AdamW | AdamW | AdamW | AdamW |
| Learning rate | 0.0001 | 0.0001 | 0.0001 | 0.0001 | 0.0001 |
| Batch size | 256 | 256 | 256 | 256 | 256 |
| Visual Encoder | – | – | DINOv2-B ($\lambda = 0.5$) | – | – |
| Token drop ratio $r$ | 75% | 75% | 75% | 75% | 75% |
| PDG drop ratio $p$ | 10% | 10% | 10% | 10% | 10% |
| *Finetuning config.* | | | | | |
| Training iterations | – | 100K | 100K | 200K | 100K |
| Warmup iterations | – | 5K | 5K | 5K | 5K |
| Optimizer | – | AdamW | AdamW | AdamW | AdamW |
| Learning rate | – | 0.0002 | 0.0002 | 0.00015 | 0.0002 |
| Batch size | – | 512/1024 | 512/1024 | 1024 | 512 |
| Token drop ratio $r$ | 75% | 75% | 75% | 75% | 75% |
| PDG drop ratio $p$ | 10% | 10% | 10% | 10% | 10% |
| *Evaluation config.* | | | | | |
| Sampler | ODE | ODE/SDE | ODE/SDE | SDE | ODE |
| Sampling steps | 50 | 50/250 | 50/250 | 250 | 50 |

location within our sparse middle block, $g_\theta$. A key consideration is that the hidden states in $g_\theta$ operate on a sparse token set of length $N'$, while the target DINOv2 features have a full sequence length of $N$. To resolve this, we simply apply the same token-dropping mask to the DINOv2 feature sequence, ensuring a one-to-one correspondence for the alignment loss. Since DINOv2 also uses a standard transformer architecture with positional encodings, aligning the corresponding tokens is straightforward.

**U-ViT (Bao et al., 2023).** U-ViT extends the Vision Transformer with a U-Net (Ho et al., 2020)-style architecture. Similar to U-Net, it stacks transformer blocks with long skip-connections between encoder and decoder stages, directly passing features from encoder to decoder. To apply SPRINT, we first conceptually decompose the U-ViT into our standard $f_\theta$, $g_\theta$, and $h_\theta$ sections while preserving all original skip-connections. We then introduce our dense residual path between $f_\theta$ and $h_\theta$ and apply token dropping to the middle section, $g_\theta$. The U-Net skip-connections remain compatible with this design. The long-range skips between the encoder and decoder are unaffected. The shorter skip-connections within the sparse middle section naturally operate on the reduced set of tokens. This allows SPRINT to be integrated cleanly without disrupting the U-ViT's core component.

## C    IMPLEMENTATION DETAILS AND HYPERPARAMETERS

### C.1    TRAINING DETAILS

We follow the model configuration of the original SiT implementation (Ma et al., 2024), with the only modification being a single linear projection layer for sparse–dense residual fusion. This adds only a marginal number of parameters, approximately 0.3% of the original model size. We use pre-computed latent vectors from raw images via Stable Diffusion (Rombach et al., 2022) and Flux (Labs, 2024b) VAEs, and, following common practice, do not apply any data augmentation. For pretraining, we train SPRINT with a batch size of 256, a learning rate of 1e-4, a fixed drop ratio of 75%, and an EMA decay rate of 0.9999, following standard configuration (Ma et al., 2024; Yu et al., 2024; Park et al., 2024b; 2025). After pre-training, we switch the middle blocks to operate on the full token set for a short fine-tuning stage for 100K iterations. We increase the batch size and the learning rate, following standard practice (Zheng et al., 2024; Krause et al., 2025). We found that applying a linear learning

---

**Algorithm 1** SPRINT Pre-training

---

**Require:** Input $\mathbf{x}_0$, Drop ratio $r$, Path-drop prob $p$, encoder $f_\theta$, middle blocks $g_\theta$, decoder $h_\theta$, condition $\mathbf{c}$

 1: **while** not converged **do**
 2:     Sample $t \sim [0, 1]$ and $\epsilon \sim \mathcal{N}(0, I)$
 3:     $\mathbf{x}_t \leftarrow (1 - t)\,\mathbf{x}_0 + t\,\epsilon$               $\triangleright \mathbf{x}_t \in \mathbb{R}^{B \times N \times C}$
 4:     $\mathbf{f}_t \leftarrow f_\theta(\mathbf{x}_t, \mathbf{c})$               $\triangleright \mathbf{f}_t \in \mathbb{R}^{B \times N \times C}$
 5:     $\mathbf{f}_t^{drop} \leftarrow \mathrm{Drop}(\mathbf{f}_t, r)$       $\triangleright \mathbf{f}_t^{drop} \in \mathbb{R}^{B \times (1-r)N \times C}$
 6:     $\mathbf{g}_t^{drop} \leftarrow g_\theta(\mathbf{f}_t^{drop}, \mathbf{c})$      $\triangleright \mathbf{g}_t^{drop} \in \mathbb{R}^{B \times (1-r)N \times C}$
 7:     $\mathbf{g}_t^{\mathrm{pad}} \leftarrow \mathrm{PadWithMask}(\mathbf{g}_t^{drop})$     $\triangleright \mathbf{g}_t^{\mathrm{pad}} \in \mathbb{R}^{B \times N \times C}$
 8:     $\mathbf{g}_t^{\mathrm{pad}} \leftarrow$ [MASK] with probability $p$     $\triangleright$ Path-drop learning
 9:     $\mathbf{h}_t \leftarrow \mathrm{Fusion}(\mathbf{f}_t, \mathbf{g}_t^{\mathrm{pad}})$     $\triangleright$ Sparse–dense residual fusion
10:     $\hat{\mathbf{v}}_t \leftarrow h_\theta(\mathbf{h}_t, \mathbf{c})$            $\triangleright \hat{\mathbf{v}}_t \in \mathbb{R}^{B \times N \times C}$
11:     $\mathcal{L}_{vel} \leftarrow \|\hat{\mathbf{v}}_t - \mathbf{v}_t\|^2$
12:     Update $\theta$ using $\nabla_\theta \mathcal{L}_{vel}$
13: **end while**
14: **return** $f_\theta, g_\theta, h_\theta$

---

**Algorithm 2** SPRINT Fine-tuning

---

**Require:** Input $\mathbf{x}_0$, Path-drop prob $p$, encoder $f_\theta$, middle blocks $g_\theta$, decoder $h_\theta$, condition $\mathbf{c}$

 1: **while** not converged **do**
 2:     Sample $t \sim [0, 1]$ and $\epsilon \sim \mathcal{N}(0, I)$
 3:     $\mathbf{x}_t \leftarrow (1 - t)\,\mathbf{x}_0 + t\,\epsilon$         $\triangleright \mathbf{x}_t \in \mathbb{R}^{B \times N \times C}$
 4:     $\mathbf{f}_t \leftarrow f_\theta(\mathbf{x}_t, \mathbf{c})$          $\triangleright \mathbf{f}_t \in \mathbb{R}^{B \times N \times C}$
 5:     $\mathbf{g}_t \leftarrow g_\theta(\mathbf{f}_t, \mathbf{c})$          $\triangleright \mathbf{g}_t \in \mathbb{R}^{B \times N \times C}$
 6:     $\mathbf{g}_t \leftarrow$ [MASK] with probability $p$     $\triangleright$ Path-drop learning
 7:     $\mathbf{h}_t \leftarrow \mathrm{Fusion}(\mathbf{f}_t, \mathbf{g}_t)$     $\triangleright$ Sparse–dense residual fusion
 8:     $\hat{\mathbf{v}}_t \leftarrow h_\theta(\mathbf{h}_t, \mathbf{c})$          $\triangleright \hat{\mathbf{v}}_t \in \mathbb{R}^{B \times N \times C}$
 9:     $\mathcal{L}_{vel} \leftarrow \|\hat{\mathbf{v}}_t - \mathbf{v}_t\|^2$
10:     Update $\theta$ using $\nabla_\theta \mathcal{L}_{vel}$
11: **end while**
12: **return** $f_\theta, g_\theta, h_\theta$

---

rate warm-up from 2e-6 to 2e-4 over the first 5K iterations stabilizes the training. During the warm-up stage, we use an EMA decay rate of 0.999, which is restored to 0.9999 afterward. For both training phases, we introduce a path-drop learning strategy to maximize the effectiveness of our path-drop guidance, in addition to the standard class-condition dropping. Specifically, following the practice in CFG training, we randomly drop the features of the sparse–deep path with a probability of 10% and replace the dropped features with mask tokens. This random dropping is performed independently of the condition dropping in CFG. To accelerate training, we adopt mixed-precision (bf16) training and apply gradient norm clipping at 1.0 during both pretraining and finetuning. Detailed hyperparameters are summarized in Table 9. All experiments are conducted on 8 NVIDIA A100 80GB GPUs.

## C.2 EVALUATION DETAILS

**Metrics.** We evaluate generation performance using several standard metrics: FDD (Stein et al., 2023) (Fréchet Distance on DINOv2), FID (Heusel et al., 2017) (Fréchet Inception Distance), IS (Salimans et al., 2016) (Inception Score), and Precision/Recall (Kynkäänniemi et al., 2019; Park & Kim, 2023). Unless otherwise specified, we follow the evaluation protocol of (Dhariwal & Nichol, 2021) and report results using 50K generated samples.

FID is the most widely used metric, measuring the feature distance between the distributions of real and generated images. It relies on the Inception-V3 network and assumes both feature distributions follow multivariate Gaussian distributions. IS also uses the Inception-V3 network but instead evaluates

---

**Algorithm 3** SPRINT Inference

---

**Require:** encoder $f_\theta$, middle blocks $g_\theta$, decoder $h_\theta$, condition $\mathbf{c}$, guidance scale $w$, sampling steps $N$, sampler $\mathcal{S}$
1: $\mathbf{x}_1 \sim \mathcal{N}(0, \mathcal{I})$
2: **for** $i = N$ **to** 1 **do**
3:      $t \leftarrow \frac{i}{N}$
4:      **if** Path-drop guidance **then**
5:          $v(\mathbf{x}_t, \emptyset) \leftarrow h_\theta(\text{Fusion}(\text{M}, f_\theta(\mathbf{x}_t, \emptyset)), \emptyset)$             ▷ Path-drop guidance
6:      **else**
7:          $v(\mathbf{x}_t, \emptyset) \leftarrow h_\theta(\text{Fusion}(g_\theta(f_\theta(\mathbf{x}_t, \mathbf{c}), \mathbf{c}), f_\theta(\mathbf{x}_t, \emptyset)), \emptyset)$     ▷ Classifier-free guidance
8:      **end if**
9:      $\tilde{v}(\mathbf{x}_t, \mathbf{c}) \leftarrow v(\mathbf{x}_t, \emptyset) + w \cdot \big(v(\mathbf{x}_t, \mathbf{c}) - v(\mathbf{x}_t, \emptyset)\big)$
10:      $\mathbf{x}_{t-\frac{1}{N}} \leftarrow \mathcal{S}(\mathbf{x}_t, \tilde{v}(\mathbf{x}_t, \mathbf{c}))$
11: **end for**
12: **return** $\mathbf{x}_0$

---

the quality and diversity of generated images by computing the KL-divergence between the marginal label distribution and the conditional label distribution predicted from logits.

FDD adopts the same formulation as FID but replaces Inception features with DINOv2 features, which provide stronger semantic alignment and robustness to noise. Notably, FDD has been shown to be more reliable for diffusion models (Stein et al., 2023; Karras et al., 2024b).

Finally, Precision (Kynkäänniemi et al., 2019) measures the fraction of generated images that are realistic, while Recall measures the fraction of the training data manifold covered by generated samples.

**Guidance scale.** We use the following formulation for guidance sampling (Ho & Salimans, 2022):

$$\tilde{v}(\mathbf{x}_t, \mathbf{c}) = v(\mathbf{x}_t, \emptyset) + w \cdot (v(\mathbf{x}_t, \mathbf{c}) - v(\mathbf{x}_t, \emptyset)), \tag{5}$$

where $w$ denotes the guidance scale. In standard Classifier-Free Guidance (CFG), the unconditional velocity $v(\mathbf{x}_t, \emptyset)$ is computed using the full model path with a null condition. In contrast, our Path-Drop Guidance (PDG) replaces the unconditional branch with a weaker network, as defined in Eq. 4.

For the results in Tables 1 and 2, we consistently use a CFG scale of 1.4 with the ODE sampler across all methods.

For Table 3, we adopt the SDE sampler (Ma et al., 2024) to compare baselines. Under this setting, we use a CFG scale of 1.35 to achieve the best FID and 2.0 to achieve the best FDD. For our PDG sampling, the optimal scales are 1.35 for FID and 1.9 for FDD.

For our model in Table 10, we use the scale of 1.35 and 1.8 for FID and FDD, respectively, for both CFG and PDG.

# D   COMPUTATION ANALYSIS

We use the SiT-XL/2 configuration for evaluating computational analysis below.

**FLOPs.** To estimate the total training FLOPs, we measure the forward-pass FLOPs over 100 iterations with a batch size of 256, average the results, and multiply by the total number of training iterations. For inference FLOPs, we sum the forward-pass FLOPs across all sampling timesteps using a batch size of 32 and report the average over both timesteps and batch size. This procedure provides a consistent and reproducible measure of computational cost across methods. Note that we report floating-point operations (FLOPs), not multiply–accumulate operations (MACs), where one MAC corresponds to approximately two FLOPs.

**Training speed.** Here, we compare the actual run-time performance of each method on Stable Diffusion VAE latents. For all token-dropping methods, we use a fixed drop rate of 75%. At

Table 10: **Comprehensive performance comparison on ImageNet** $512 \times 512$ class-conditioned generation with classifier-free guidance. $\downarrow$ / $\uparrow$ indicate whether lower or higher values are better, respectively. All metrics are evaluated with 250 sampling steps using the SDE sampler. Training and inference TFLOPs are measured with the DeepSpeed library.

| Method | Epochs | #Params. | Training TFLOPs $\downarrow$ ($\times 10^6$) | Inference TFLOPs $\downarrow$ | FDD $\downarrow$ | FID $\downarrow$ | Pre. $\uparrow$ | Rec. $\uparrow$ |
|---|---|---|---|---|---|---|---|---|
| ADM (Dhariwal & Nichol, 2021) | 400 | – | – | – | – | 2.85 | 0.84 | 0.53 |
| Simple diffusion (U-Net) | 800 | – | – | – | – | 4.28 | – | – |
| Simple diffusion (U-ViT-L) | 800 | – | – | – | – | 4.53 | – | – |
| MaskDiT (Zheng et al., 2024) | 800 | 730M | 327.2 | 1.029 | – | 2.50 | 0.83 | 0.56 |
| DiT-XL (Peebles & Xie, 2023) | 600 | 675M | 366.6 | 0.952 | – | 3.04 | 0.84 | 0.54 |
| SiT-XL (Ma et al., 2024) | 600 | 675M | 366.6 | 0.952 | – | 2.62 | 0.84 | 0.57 |
| SiT-XL | | | | | | | | |
| + **SPRINT** | 400 | 677M | **184.8** | 0.954 | 53.6 | 2.23 | 0.83 | 0.57 |
| + **SPRINT**$_{PDG}$ | 400 | 677M | **184.8** | **0.471** | 46.9 | **1.96** | 0.83 | 0.58 |

the ImageNet resolution of $256^2$, SPRINT achieves a pretraining speed of **5.2** iters/sec, which is more than **2× faster** than the SiT baseline (2.5 iters/sec) and clearly outperforms other token-dropping baselines, including MaskDiT (4.57 iters/sec), MicroDiT (3.9 iters/sec), and Tread (4.7 iters/sec). At the higher ImageNet resolution of $512^2$, SPRINT maintains its advantage, achieving **2.01** iters/sec—over **2.5× faster** than the SiT baseline (0.79 iters/sec)—and again surpassing MaskDiT (1.77 iters/sec), MicroDiT (1.54 iters/sec), and Tread (1.79 iters/sec). This acceleration results in substantial reductions in wall-clock training time and GPU consumption, making large-scale diffusion model training significantly more practical and resource-efficient.

**VRAM memory consumption.** In addition to reducing computational cost, SPRINT significantly lowers GPU memory requirements during training. For example, when training with a batch size of 32 and image resolution $256^2$ on a single GPU, SPRINT requires only 19.6 GB of memory, compared to 29.6 GB for the baseline SiT-XL/2 model. At resolution $512^2$, our SPRINT requires 37.9 GB, whereas the baseline SiT-XL/2 model requires 77.7 GB. This represents a **33.8% reduction** in memory usage at $256^2$ and a **51.2% reduction** at $512^2$. Such efficiency enables training with larger batch sizes or higher resolutions on the same hardware, making our method more accessible for researchers with limited GPU resources. Importantly, this reduction comes without sacrificing performance, underscoring the practicality of SPRINT in resource-constrained environments.

# E BASELINES

## E.1 BASELINE DETAILS ON TABLE 1

For a fair system-level comparison in Tab. 1, we apply the *same pretraining and finetuning strategies*, along with identical transformer block configurations, a fixed drop ratio of 75%, and consistent evaluation hyperparameters, across all baselines.

**Progressive training.** We adopt the same network architecture for progressive training. The model is first pretrained on $128 \times 128$ images and then finetuned on $256 \times 256$ images, with positional embeddings resized using bilinear interpolation during the resolution transition. This approach is slightly more efficient than SPRINT in terms of computational cost per iteration, achieving 25.8 vs. 31.5 GFLOPs ($\times 10^9$) at 1M training iterations. However, despite the efficiency advantage, progressive training lags behind SPRINT in performance and even fails to match the baseline SiT results, underscoring its limited effectiveness.

**MicroDiT (Sehwag et al., 2025).** MicroDiT introduces deferred masking, where token dropping is applied only after several additional patch-mixing blocks. These modules allow local patch tokens to fuse information, enriching their semantic content. Following the original protocol, we modify the SiT-XL/2 model by inserting patch-mixing modules composed of six transformer blocks. As shown in Tab. 1, this modification substantially increases computational cost and the number of parameters. Nevertheless, despite the additional overhead, MicroDiT underperforms relative to

SPRINT, highlighting that the deferred masking strategy and additional compute does not translate into superior efficiency or accuracy.

**Tread (Krause et al., 2025).** Tread introduces a token-routing strategy in which randomly dropped tokens at early layers are routed directly to deeper layers. While this resembles SPRINT in that tokens bypass the middle layers, the two approaches differ fundamentally. In Tread, only the dropped tokens are bypassed, forcing the middle block to encode local noise information in order to estimate velocity. In contrast, SPRINT employs a full dense residual path that delivers complete local noise information to the decoder, freeing the middle block to focus on modeling global contextual information. This design choice makes SPRINT *highly effective under aggressive dropping ratios* (75%), whereas Tread fails under the same setting. We follow the implementation details provided in the original Tread paper.

### E.2 MORE DISCUSSION ON OTHER BASELINES

**MaskDiT (Zheng et al., 2024).** MaskDiT introduces an additional reconstruction task for masked tokens alongside the diffusion objective, encouraging the model to recover missing information and thereby improve contextual understanding. While this approach provides some efficiency gains, it requires an extra decoder module, increasing the model size from 675M to 730M and adding computational overhead. Moreover, its effectiveness is limited to moderate dropping ratios (e.g., 50%). As shown in Tab. 3, these limitations restrict its overall efficiency compared to our framework. Specifically, MaskDiT requires 1600 training epochs to reach 65.4 FDD and 2.28 FID, whereas SPRINT surpasses this in just 200 epochs with 61.8 FDD and 2.01 FID. This underscores the superior effectiveness and efficiency of SPRINT over MaskDiT.

**MDT (Gao et al., 2023).** The Masked Diffusion Transformer (MDT) also aims to improve the contextual understanding of diffusion models through token dropping. They designed masked diffusion transformer with encoder-decoder split of the diffusion transformer, where the encoder processes masked tokens and forwards them to the decoder along with remaining tokens through additional side-interpolator model. It adds additional long shorcut connections between encoder blocks along with long full token input to all decoder blocks. MDT optimizes the reconstruction loss on masked tokens along with diffusion loss. The added complexity in the training and architectural changes is aimed for better generative performance. Similar to MaskDiT, this work also operates only with moderate token dropping ratios (e.g., [30%, 50%]). MDT does not work well with high token dropping ratio such as 75%.

## F ADDITIONAL QUANTITATIVE RESULTS

### F.1 IMAGENET 512x512 EXPERIMENT

In the main text, we have already demonstrate that SPRINT outperforms many existing training methods and state-of-the-art models at $256^2$ class conditional image generation. In this experiment, we train our models to generation images at $512^2$ resolution.

Tab. 10 compares our method with strong baselines on ImageNet-1K class-conditional generation at $512^2$. We pre-train SPRINT for 1.8M iterations and finetune for 200K iterations (refer to Table 9). SPRINT achieves comparable or better generation quality while using substantially fewer training TFLOPs ($\times 10^6$): only 184.8 at 400 epochs, versus 366.6 for SiT-XL at 600 epochs. This demonstrates much faster convergence, reaching better FID with nearly $2\times$ lower training cost. At inference, Path-Drop Guidance provides further benefits, nearly halving inference TFLOPs (0.471 vs. 0.952) while improving FDD. Overall, SPRINT consistently demonstrates efficiency compared to the baselines at $512^2$, by combining lower training and inference costs. Refer to Fig. 33 for qualitative results.

### F.2 PERFORMANCE WITH FEW-STEP GENERATION

Tab. 11 compares SiT-XL/2 and SiT-XL/2 + SPRINT across lower inference steps (NFEs), an essential setting for achieving efficient and practical image generation. In real-world scenarios, reducing the number of function evaluations (NFEs) directly translates to faster sampling and lower inference

Table 11: **Performance of SiT-XL/2 and SPRINT across NFEs.** Results are reported at 1M training iterations using the ODE sampler with 50K generated samples.

| Method | NFE | FDD ↓ | FID ↓ | IS ↑ | Pre. ↑ | Rec. ↑ |
|---|---|---|---|---|---|---|
| SiT-XL/2 | 200 | 132.3 | 2.18 | 249.9 | 0.81 | 0.59 |
| + **SPRINT** (Ours) | 200 | **120.4** | **2.08** | **272.2** | 0.81 | **0.60** |
| **Gain** △ | | **+11.9** | **+0.1** | **+22.3** | | |
| SiT-XL/2 | 150 | 133.1 | 2.19 | 249.6 | 0.81 | 0.59 |
| + **SPRINT** (Ours) | 150 | **121.1** | **2.09** | **271.5** | 0.81 | 0.59 |
| **Gain** △ | | **+12.0** | **+0.1** | **+21.9** | | |
| SiT-XL/2 | 100 | 134.7 | 2.22 | 248.4 | 0.81 | 0.58 |
| + **SPRINT** (Ours) | 100 | **122.2** | **2.10** | **271.0** | 0.81 | **0.59** |
| **Gain** △ | | **+12.4** | **+0.12** | **+22.6** | | |
| SiT-XL/2 | 50 | 140.6 | 2.34 | 244.0 | 0.80 | 0.58 |
| + **SPRINT** (Ours) | 50 | **126.5** | **2.19** | **267.7** | **0.81** | **0.59** |
| **Gain** △ | | **+14.1** | **+0.15** | **+23.7** | | |
| SiT-XL/2 | 25 | 156.1 | 2.91 | 234.4 | 0.80 | 0.57 |
| + **SPRINT** (Ours) | 25 | **138.2** | **2.59** | **256.3** | 0.80 | **0.58** |
| **Gain** △ | | **+17.9** | **+0.32** | **+21.9** | | |
| SiT-XL/2 | 10 | 222.4 | 7.37 | 187.3 | 0.74 | 0.54 |
| + **SPRINT** (Ours) | 10 | **191.7** | **6.29** | **211.3** | 0.74 | 0.54 |
| **Gain** △ | | **+30.7** | **+1.08** | **+24.0** | | |

cost, often at the expense of generation quality. While both models perform similarly at large NFEs (200), SPRINT consistently outperforms the baseline as the number of steps decreases. At 50 steps, SPRINT improves FID from 2.34 to 2.19 and IS from 244.0 to 267.7, and at only 10 steps it achieves a much larger gain, reducing FID from 7.37 to 6.29 and improving IS from 187.3 to 211.3. These results highlight that SPRINT is more competitive under low-step inference. This demonstrates the *strong representational power* of fused dense–shallow and sparse–deep features.

# G ADDITIONAL QUALITATIVE RESULTS

## G.1 VISUAL COMPARISON ON IMAGENET $256 \times 256$

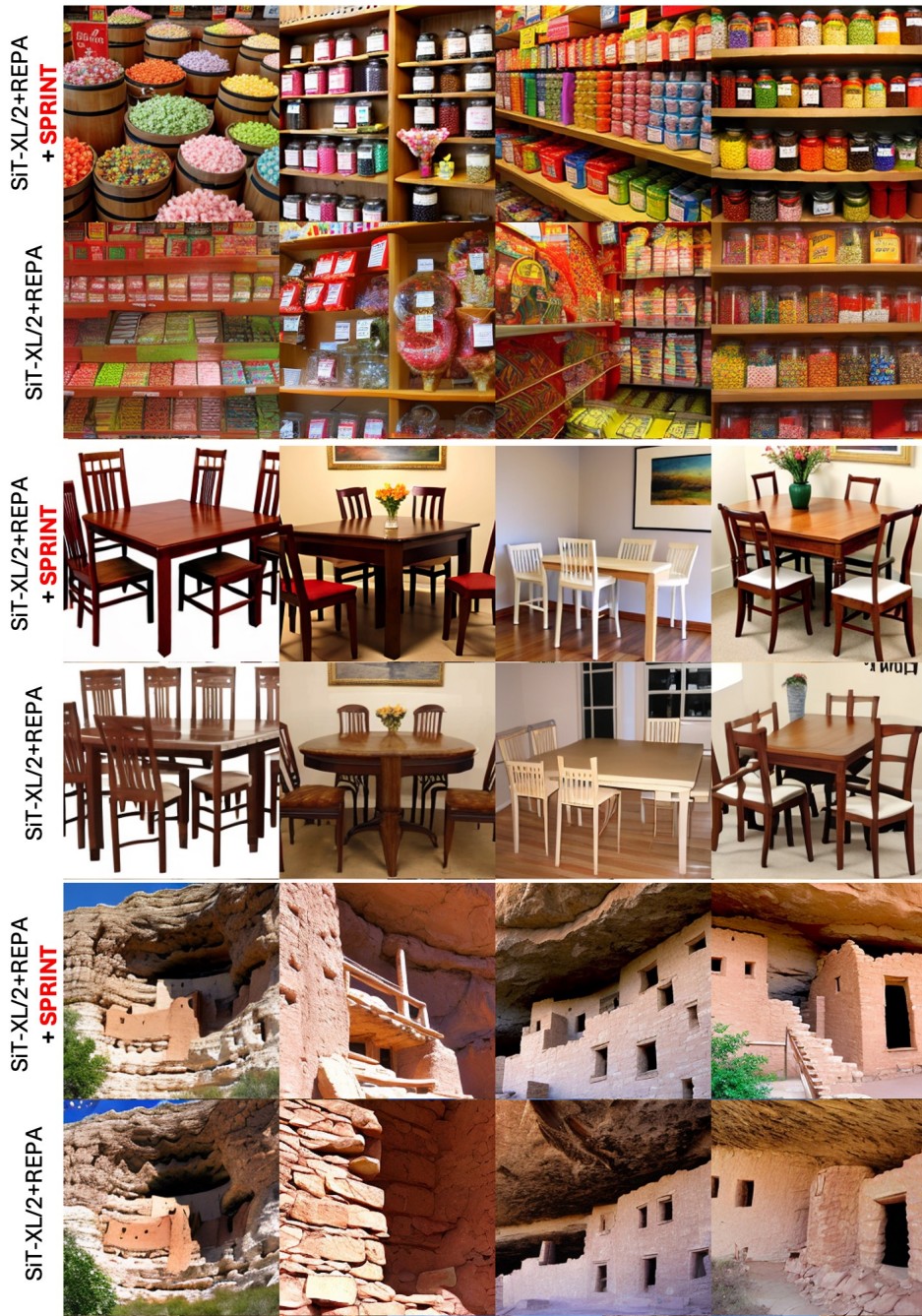

Figure 10: **SPRINT improves visual quality over baseline with only 57% of inference FLOPs (additional examples)**. We present samples from two SiT-XL/2 + REPA models after 1M training iterations, where SPRINT is applied to one of the models. For our approach, we further incorporate the proposed Path-Drop Guidance (PDG), yielding higher visual quality compared to the REPA.

## G.2 UNSELECTED GENERATED RESULTS BY SPRINT ON IMAGENET $256 \times 256$

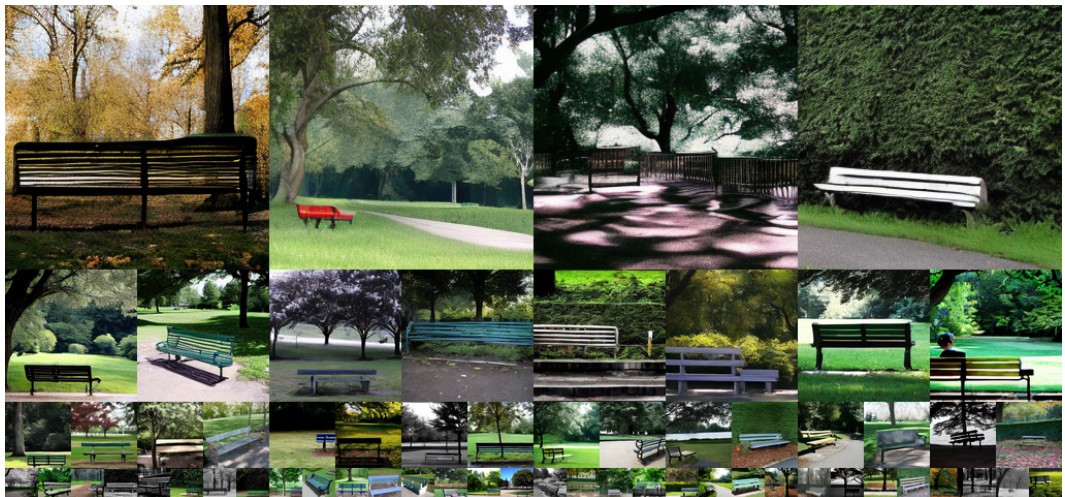

Figure 11: **Unselected generation results of SiT-XL/2 + SPRINT_CFG.** We use classifier-free guidance with w = 4.0. Class label = "park bench" (706)

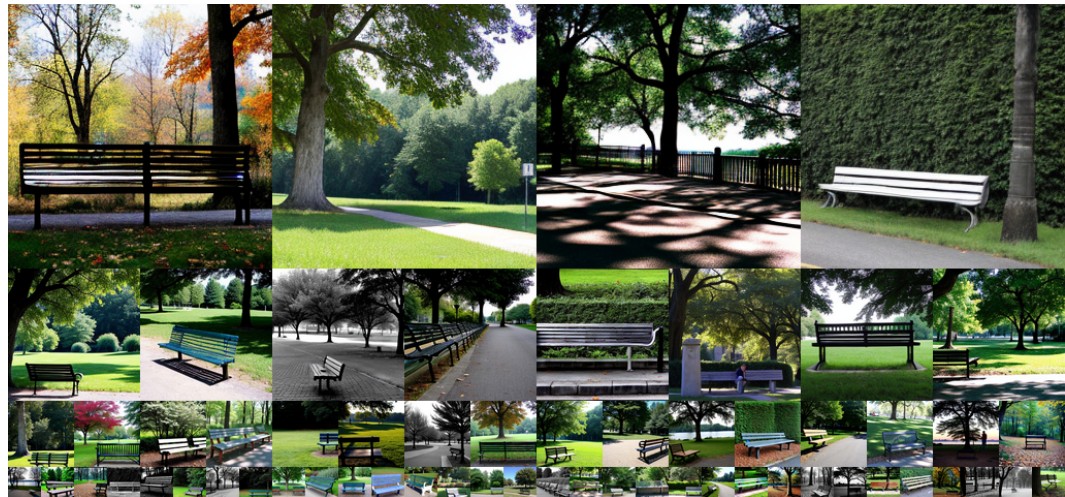

Figure 12: **Unselected generation results of SiT-XL/2 + SPRINT_PDG.** We use our path-drop guidance with w = 4.0. Class label = "park bench" (706)

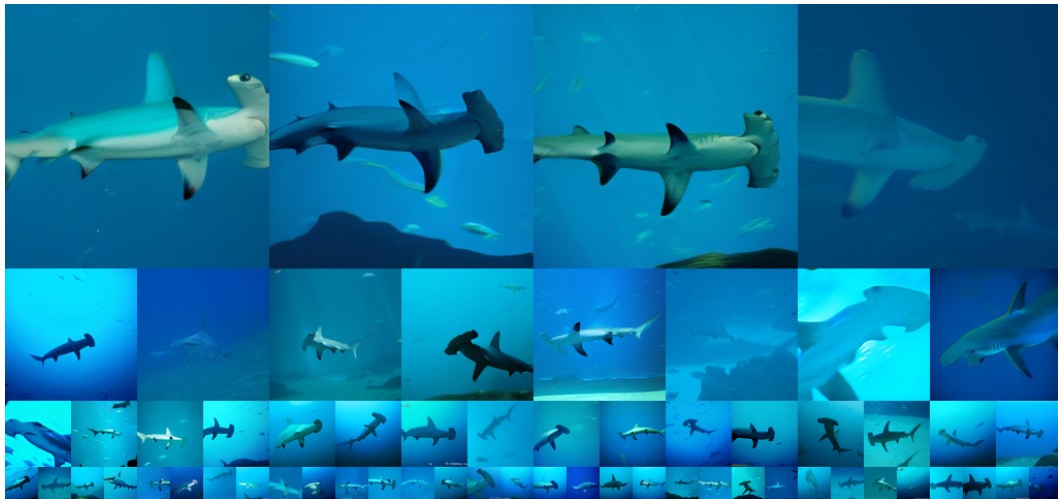

Figure 13: **Unselected generation results of SiT-XL/2 + SPRINT_CFG.** We use classifier-free guidance with w = 4.0. Class label = "hammerhead, hammerhead shark" (4)

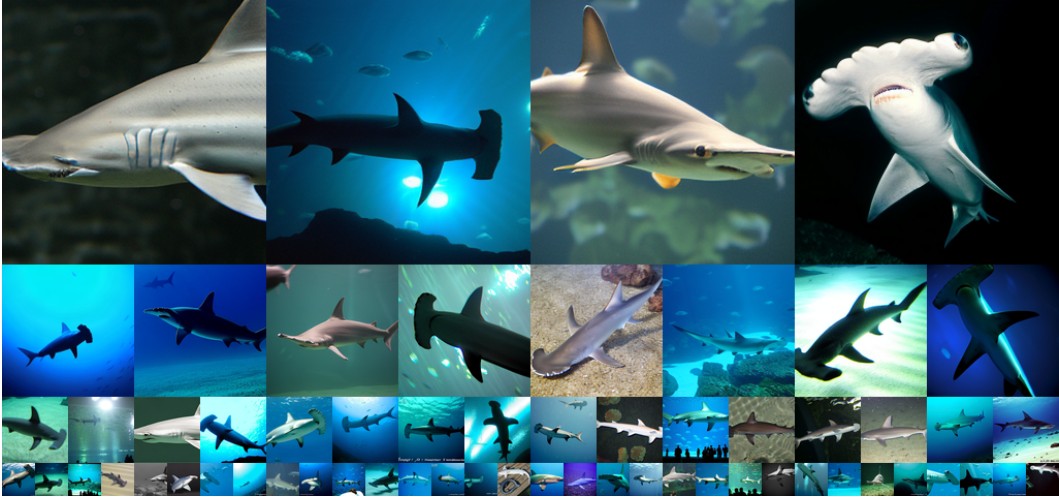

Figure 14: **Unselected generation results of SiT-XL/2 + SPRINT_PDG.** We use our path-drop guidance with w = 4.0. Class label = "hammerhead, hammerhead shark" (4)

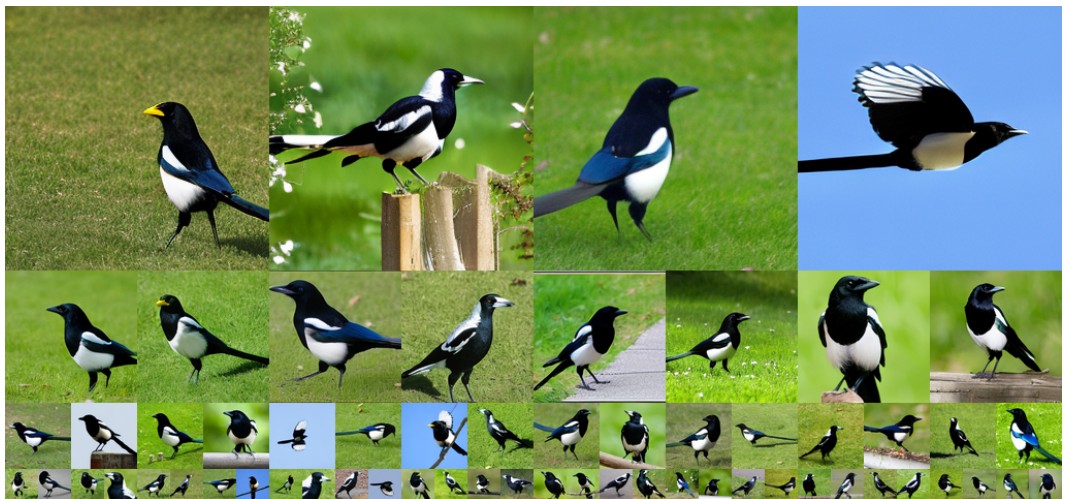

Figure 15: **Unselected generation results of SiT-XL/2 + SPRINT_CFG.** We use classifier-free guidance with w = 4.0. Class label = "magpie" (18)

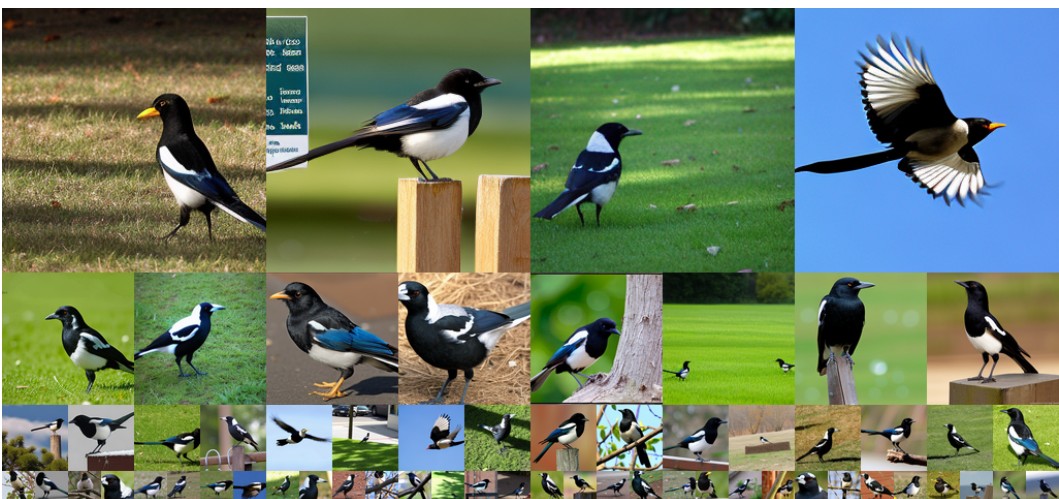

Figure 16: **Unselected generation results of SiT-XL/2 + SPRINT_PDG.** We use our path-drop guidance with w = 4.0. Class label = "magpie" (18)

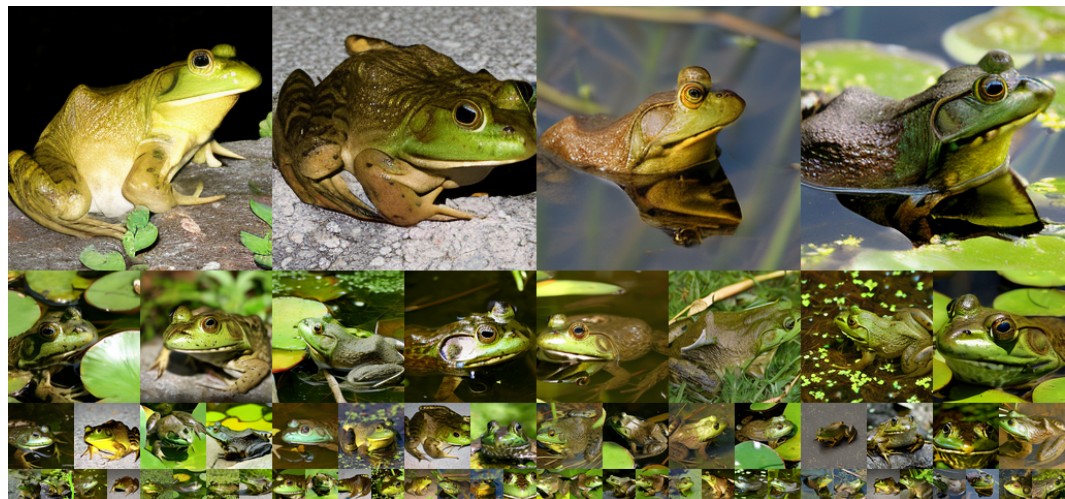

Figure 17: **Unselected generation results of SiT-XL/2 + SPRINTCFG.** We use classifier-free guidance with w = 4.0. Class label = "bullfrog, Rana catesbeiana" (30)

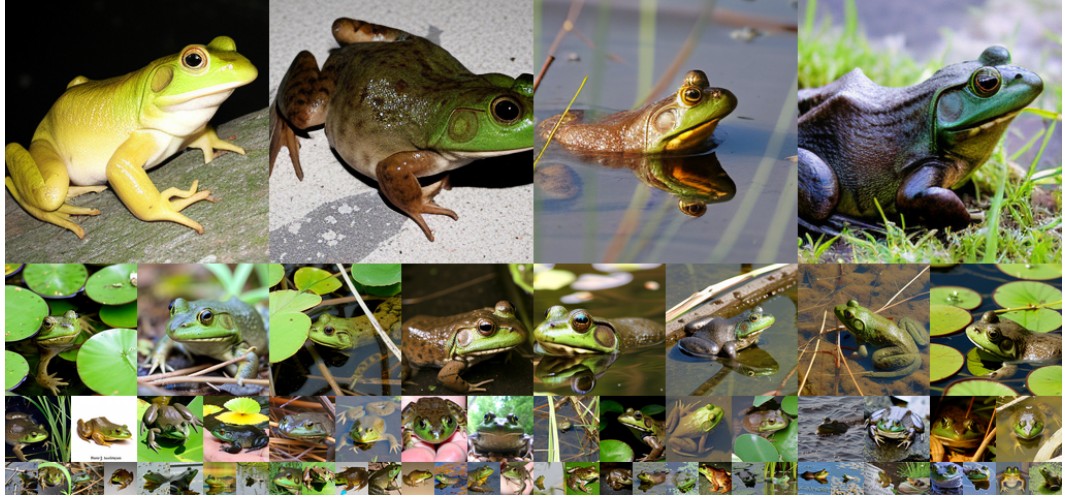

Figure 18: **Unselected generation results of SiT-XL/2 + SPRINTPDG.** We use our path-drop guidance with w = 4.0. Class label = "bullfrog, Rana catesbeiana" (30)

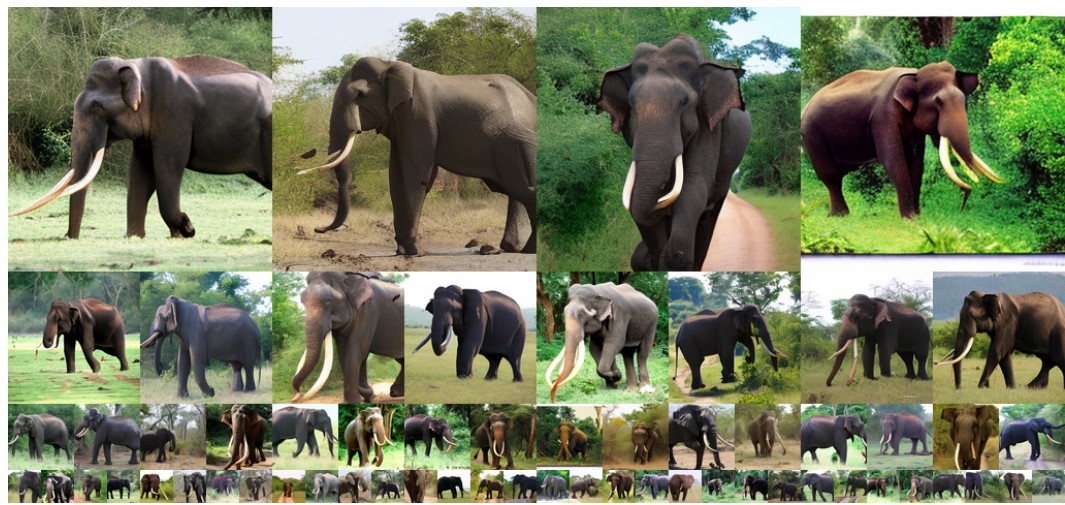

Figure 19: **Unselected generation results of SiT-XL/2 + SPRINT_CFG.** We use classifier-free guidance with w = 4.0. Class label = "tusker" (101)

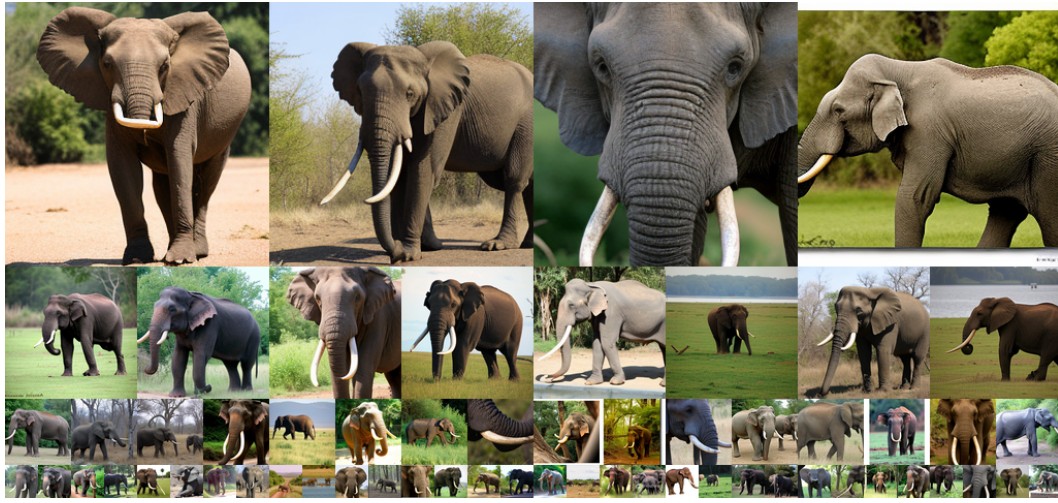

Figure 20: **Unselected generation results of SiT-XL/2 + SPRINT_PDG.** We use our path-drop guidance with w = 4.0. Class label = "tusker" (101)

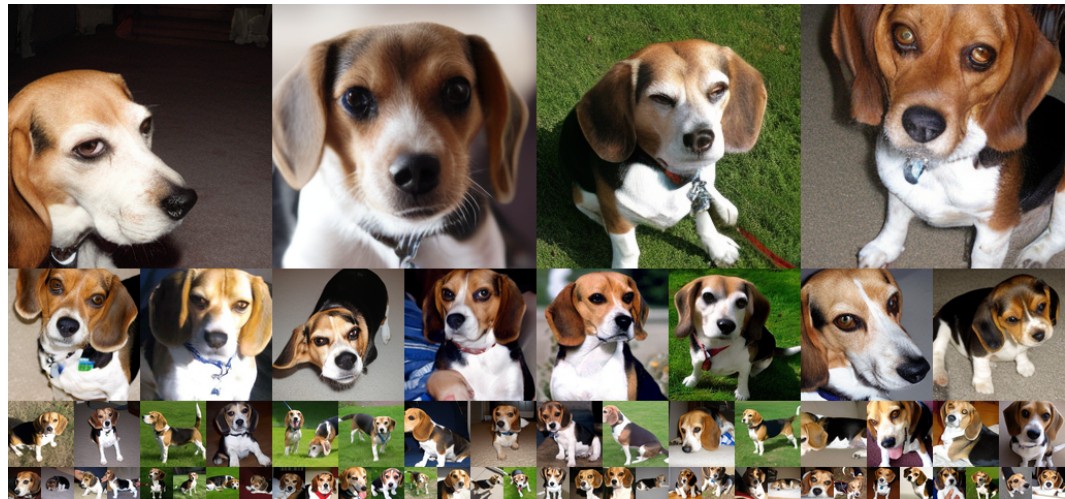

Figure 21: **Unselected generation results of SiT-XL/2 + SPRINT_CFG.** We use classifier-free guidance with w = 4.0. Class label = "beagle" (162)

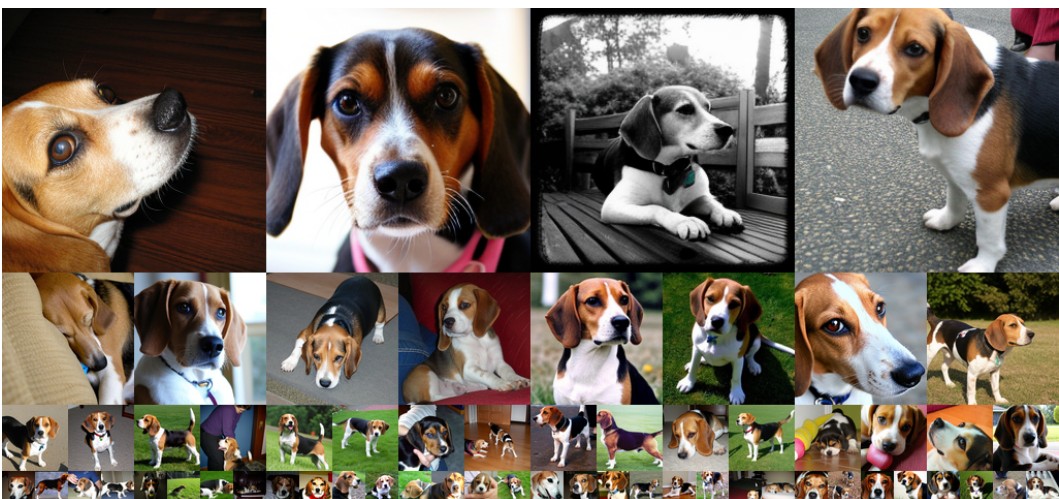

Figure 22: **Unselected generation results of SiT-XL/2 + SPRINT_PDG.** We use our path-drop guidance with w = 4.0. Class label = "beagle" (162)

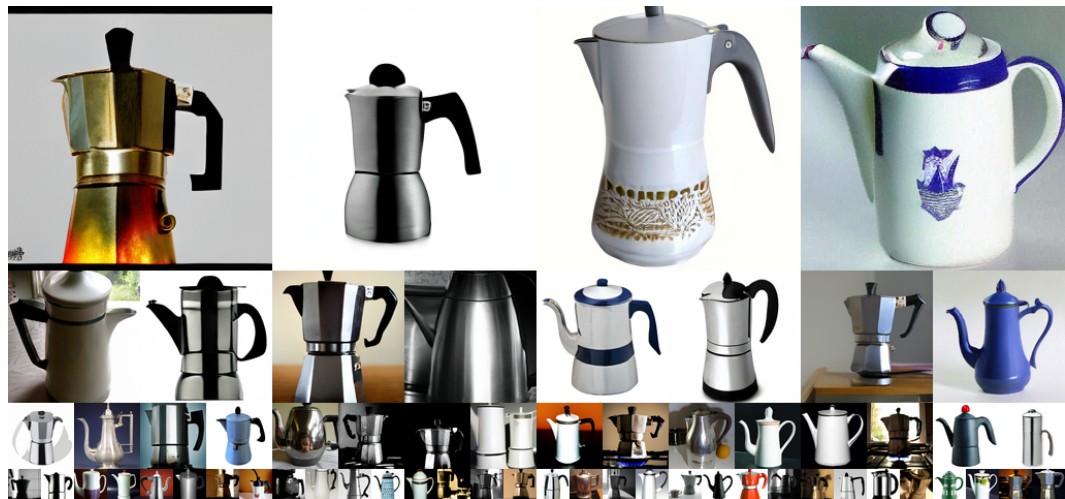

Figure 23: **Unselected generation results of SiT-XL/2 + SPRINT_CFG.** We use classifier-free guidance with w = 4.0. Class label = "coffeepot" (505)

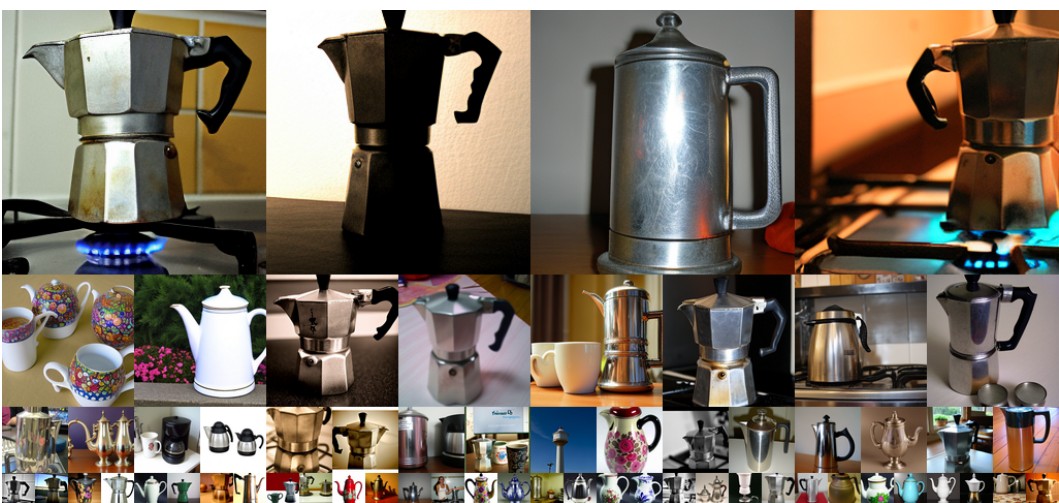

Figure 24: **Unselected generation results of SiT-XL/2 + SPRINT_PDG.** We use our path-drop guidance with w = 4.0. Class label = "coffeepot" (505)

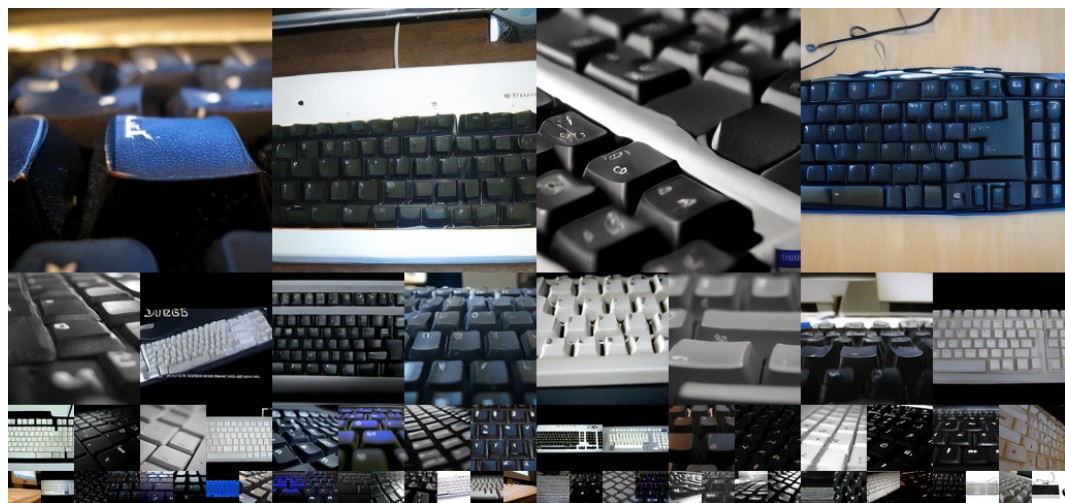

Figure 25: **Unselected generation results of SiT-XL/2 + SPRINT_{CFG}.** We use classifier-free guidance with w = 4.0. Class label = "computer keyboard, keypad" (508)

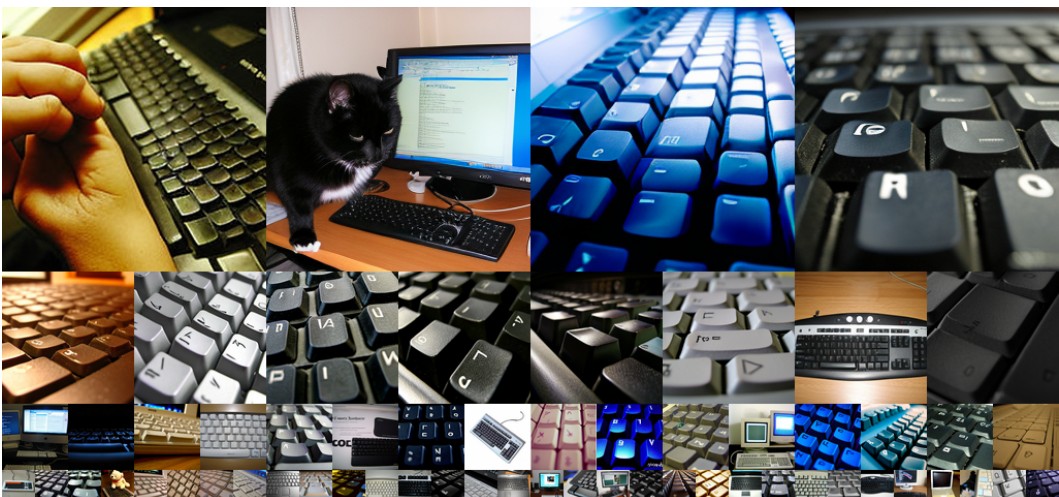

Figure 26: **Unselected generation results of SiT-XL/2 + SPRINT_{PDG}.** We use our path-drop guidance with w = 4.0. Class label = "computer keyboard, keypad" (508)

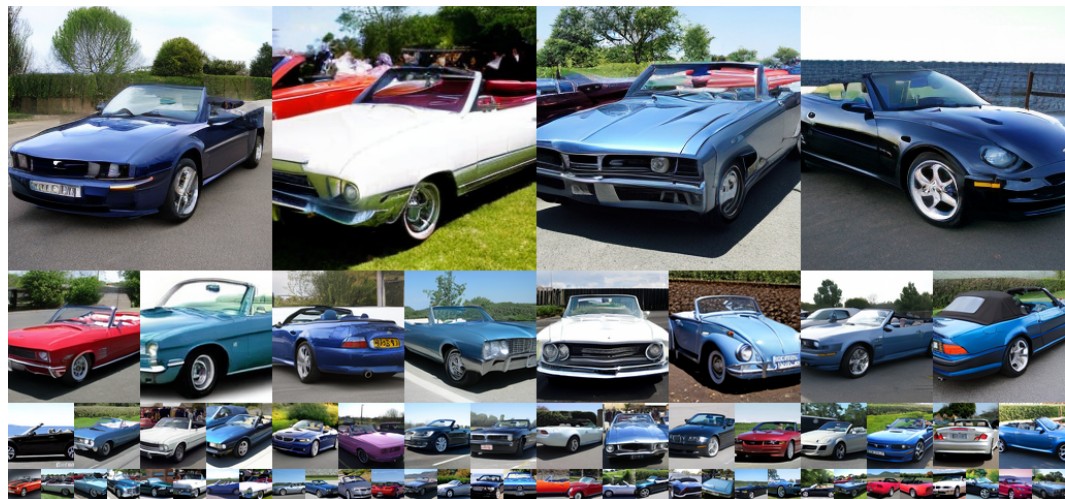

Figure 27: **Unselected generation results of SiT-XL/2 + SPRINT_CFG.** We use classifier-free guidance with w = 4.0. Class label = "convertible" (511)

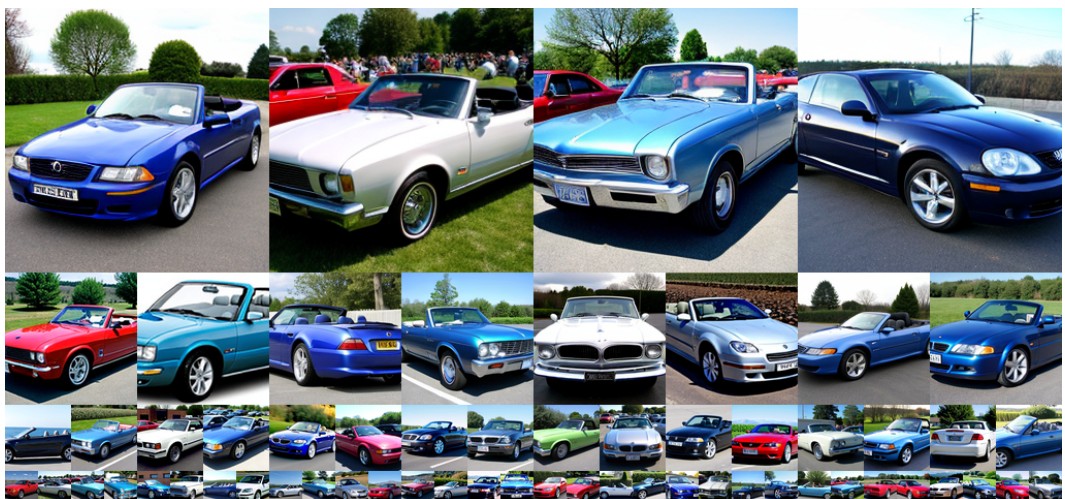

Figure 28: **Unselected generation results of SiT-XL/2 + SPRINT_PDG.** We use our path-drop guidance with w = 4.0. Class label = "convertible" (511)

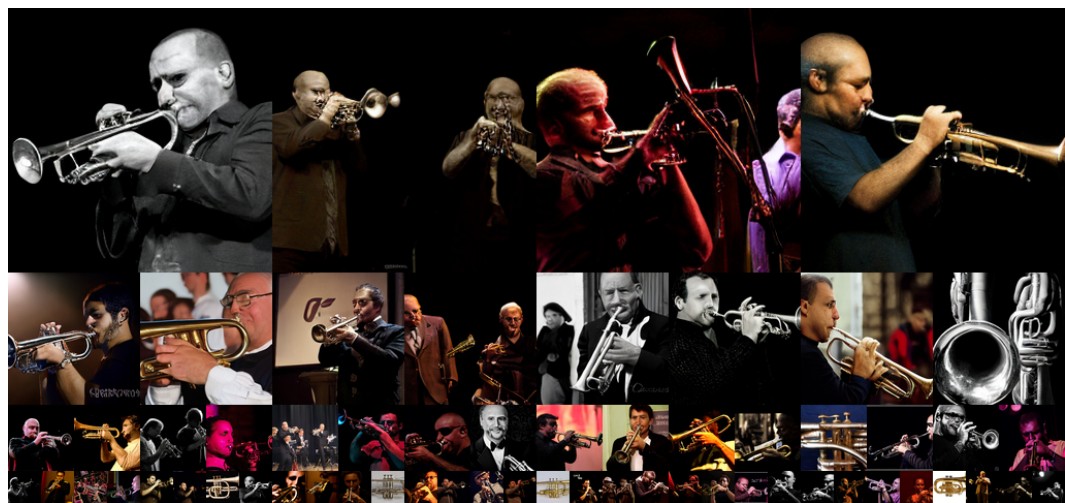

Figure 29: **Unselected generation results of SiT-XL/2 + SPRINT_{CFG}.** We use classifier-free guidance with w = 4.0. Class label = "cornet, horn, trumpet, trump" (513)

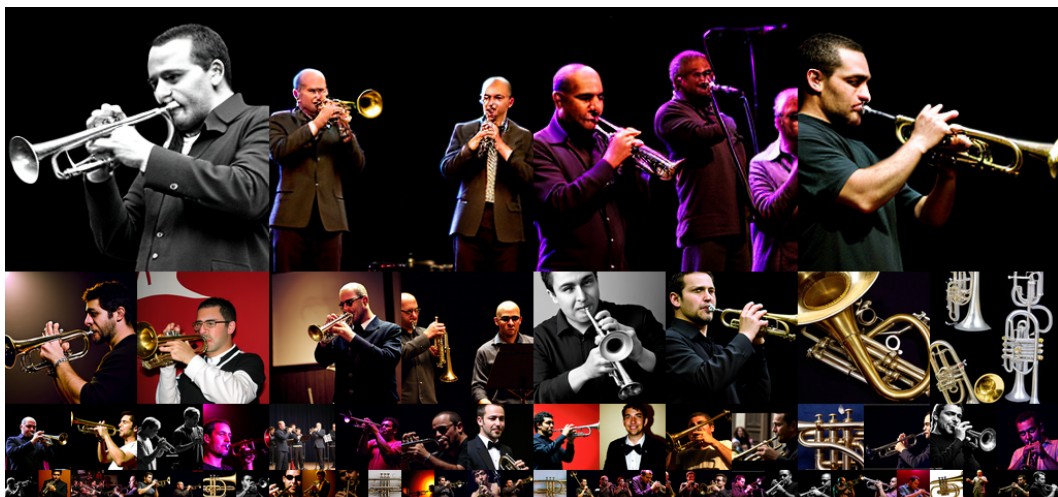

Figure 30: **Unselected generation results of SiT-XL/2 + SPRINT_{PDG}.** We use our path-drop guidance with w = 4.0. Class label = "cornet, horn, trumpet, trump" (513)

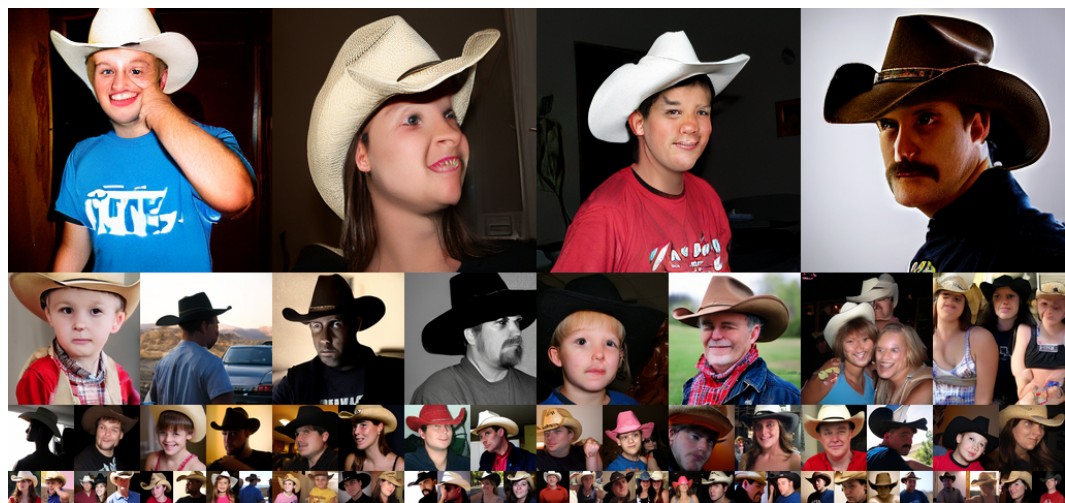

Figure 31: **Unselected generation results of SiT-XL/2 + SPRINT_CFG.** We use classifier-free guidance with w = 4.0. Class label = "cowboy hat, ten-gallon hat" (515)

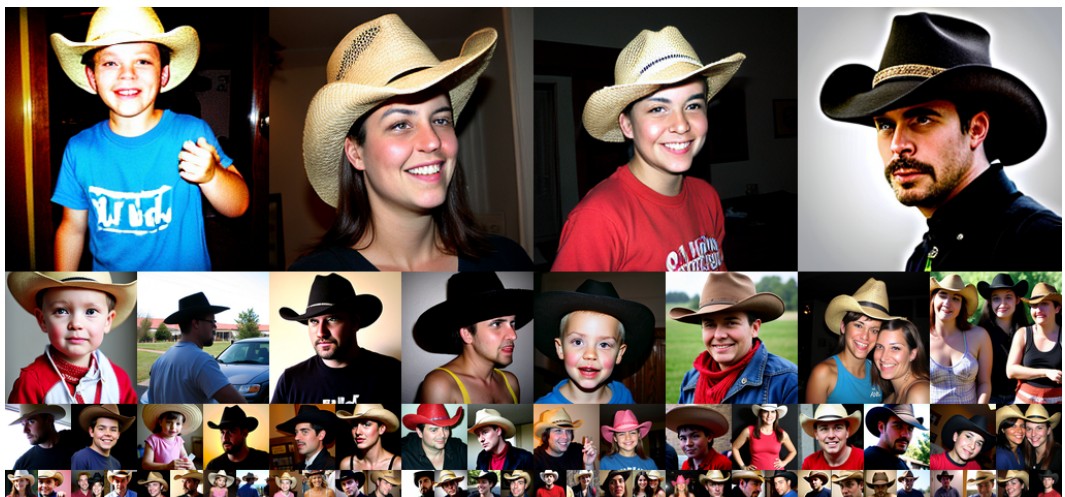

Figure 32: **Unselected generation results of SiT-XL/2 + SPRINT_PDG.** We use our path-drop guidance with w = 4.0. Class label = "cowboy hat, ten-gallon hat" (515)

## G.3   GENERATED RESULTS BY SPRINT ON IMAGENET $512 \times 512$

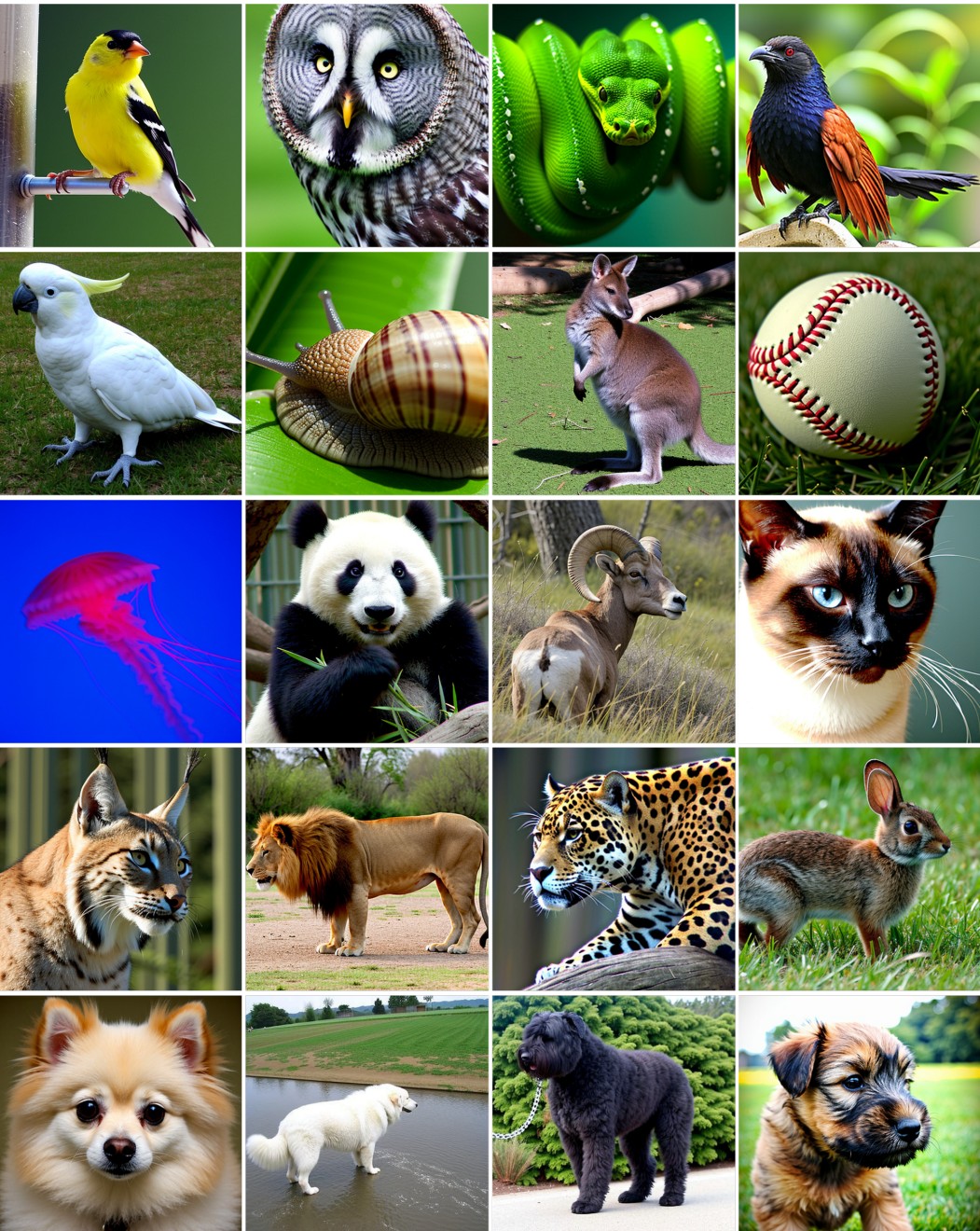

Figure 33: **Generation results of SiT-XL/2 + SPRINT_PDG.** We use our path-drop guidance with w = 3.0.

### G.4 ADDITIONAL FEATURE PCA VISUALIZATION

In the main text (Figure 6), we analyzed PCA visualizations of features from $f_\theta$ and $g_\theta$, showing that SPRINT learns more noise-invariant and semantically vivid representations than the SiT baseline across diffusion timesteps. Figure 34 presents additional examples of these dense–shallow and sparse–deep features learned by SPRINT, contrasted with those from a standard SiT-XL/2 model trained with full tokens.

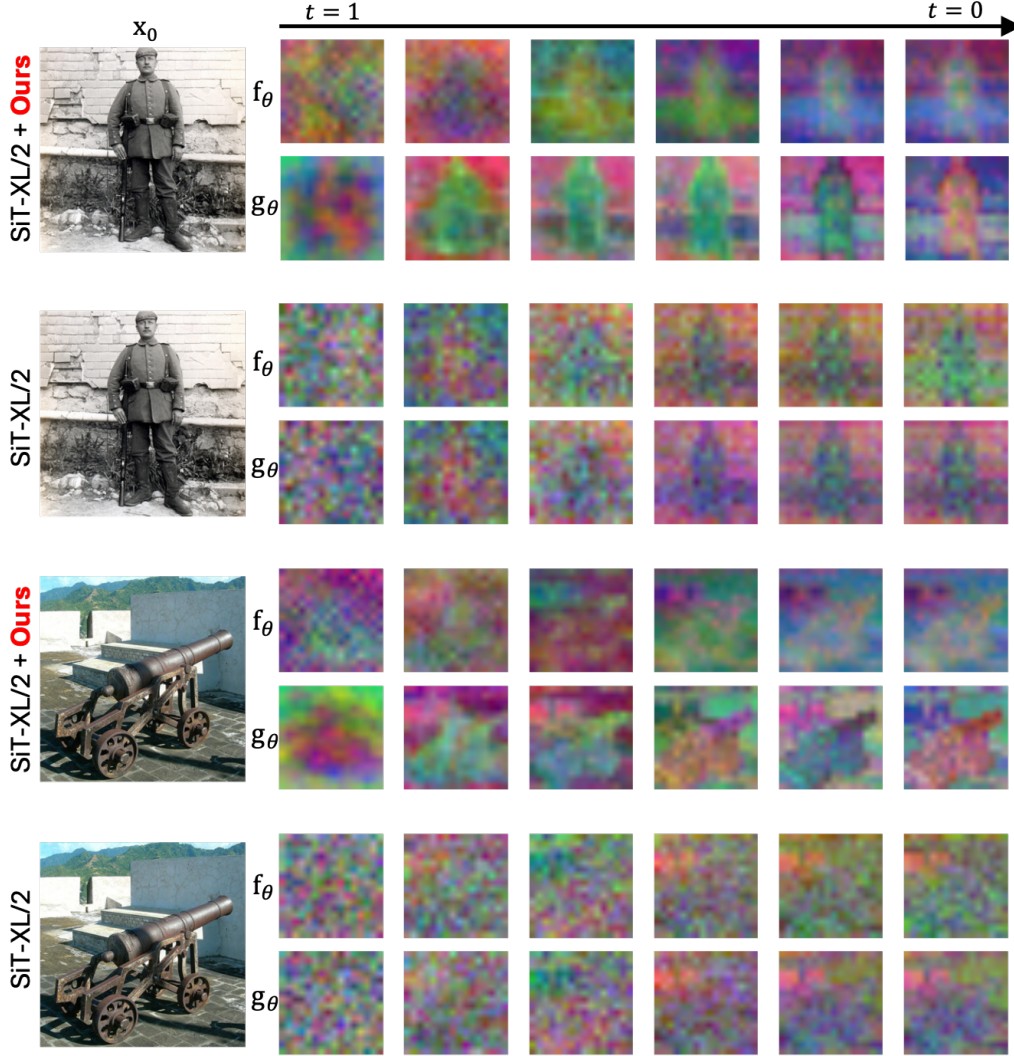

Figure 34: **SPRINT improves feature semantics (additional examples).** We visualize PCA features of $f_\theta$ and $g_\theta$ from two SiT-XL/2 models at 400K iterations. The top rows show the model trained with SPRINT, while the bottom rows show the baseline. Compared to the baseline, features from SPRINT exhibit clearer semantic structure across both images.

## H Limitation and Future Work

Our study is limited by the available computational resources, which prevented us from conducting experiments on large-scale text-to-image or video diffusion models. Exploring the scalability of SPRINT in such settings remains an important direction. In particular, the quadratic complexity of transformers becomes increasingly prohibitive as model size and input resolution grow. Since SPRINT is specifically designed to reduce redundant computation in deeper layers, we expect it to be especially beneficial for large-scale architectures where efficiency bottlenecks are most severe. Thus, extending SPRINT to other modalities such as video, 3D, or multi-modal generative models is an exciting direction. These domains pose even greater computational and memory challenges, particularly in video, where the temporal dimension compounds complexity, making our sparse–dense residual fusion especially relevant for future research.

Another promising avenue is the integration of SPRINT with recent advances in efficient attention mechanisms and scalable training strategies. Such combinations could amplify the benefits of our approach, further reducing training and inference costs while maintaining or improving performance.

