# OpenReview forum: "SPRINT: Sparse-Dense Residual Fusion for Efficient Diffusion Transformers"
_ICLR.cc/2026/Conference — ICLR 2026 Poster_

### Official Review · Reviewer_VDGb · 2025-10-30

**Soundness:** 4
**Presentation:** 3
**Contribution:** 4
**Rating:** 8
**Confidence:** 2

**Summary:**

This paper addresses the prohibitive quadratic training cost of Diffusion Transformers (DiTs). The authors propose SPRINT (Sparse-Dense Residual Fusion), a novel training and architecture strategy that enables aggressive token dropping (up to 75%) while preserving quality. SPRINT divides the DiT into three parts: a dense shallow encoder (fθ) that processes all tokens to capture local detail, a sparse deep middle block (gθ) that processes a subset of tokens to model global semantics, and a decoder (hθ) that fuses features from both paths via a residual connection. The method uses a two-stage training schedule (long sparse pre-training followed by short full-token fine-tuning) and a structured group-wise token sampling strategy to ensure local coverage. Additionally, the paper introduces Path-Drop Guidance (PDG), an efficient substitute for Classifier-Free Guidance (CFG) that nearly halves inference FLOPs by bypassing the expensive middle blocks during the unconditional pass. Experiments on ImageNet show SPRINT achieves significant training speedups (e.g., 9.8x fewer TFLOPs to reach comparable quality to SiT-XL) and improved inference efficiency with PDG.

**Strengths:**

1. Significant Training Efficiency: The primary strength is the massive reduction in training computation. The paper demonstrates that SPRINT can reach a quality level comparable to the 1400-epoch SiT-XL baseline in only 200 epochs, translating to a 9.8x reduction in TFLOPs. This is a highly practical and valuable contribution.
 2. Novel and Effective Inference Acceleration (PDG): The proposed Path-Drop Guidance (PDG) is an excellent finding. It cleverly repurposes the SPRINT architecture to create an efficient alternative to CFG, using the shallow path as a ""weaker network"". This method nearly halves inference FLOPs while also improving generation quality, as shown in Tables 1, 2, 3, and 10, and Figure 7.
 3. Simplicity and Generality: The SPRINT framework is conceptually simple and not tied to a specific architecture. The authors demonstrate its broad applicability by successfully integrating it with SiT, U-ViT, and the alignment-based REPA, showing consistent and significant improvements in all cases.
 4. Strong Empirical Validation: The paper is supported by extensive experiments and insightful ablations. The validation of the sparse-dense design (Table 4), the structured sampling strategy (Table 5), and the optimal layer allocations (Tables 6, 8) builds strong confidence in the method's design. The analysis of feature specialization (Fig. 4) provides good intuition for why the method works.

**Weaknesses:**

1. Limited Ablation on Fine-tuning Stage: The paper adopts a fixed two-stage schedule: long sparse pre-training followed by a 100K-iteration full-token fine-tuning stage. While effective, the sensitivity to the duration of this fine-tuning is not analyzed. It is unclear how much fine-tuning is necessary to close the train-inference gap or if more fine-tuning would yield further gains. An ablation on this hyperparameter would strengthen the paper.
 2. Conceptual Justification for PDG: The paper empirically shows that PDG works well by using the shallow path fθ as the unconditional estimate, inspired by Auto Guidance. This is effective, but a deeper conceptual analysis of why fθ is a suitable unconditional predictor would be beneficial. Does the SPRINT training force fθ to learn an ""average"" or ""blurry"" representation that mimics a true unconditional pass? A brief discussion would be insightful.

**Questions:**

1.The performance of PDG is excellent. Does the SPRINT training enable PDG? In other words, could PDG (i.e., using only the first few layers fθ for the unconditional pass) be applied as an inference-time optimization to a standard, densely trained SiT model, or is the sparse-dense training process necessary for fθ to become an effective unconditional predictor?
2. Could you provide experimental results on text-to-image models (such as PixArt, Flux), similar to the fine-tuning experiments on FLUX, to more convincingly demonstrate the effectiveness of your proposed method?

---

> ### Author Response · Authors · 2025-11-19
>
> >***Q1. Limited Ablation on Fine-tuning Stage: The paper adopts a fixed two-stage schedule: long sparse pre-training followed by a 100K-iteration full-token fine-tuning stage. While effective, the sensitivity to the duration of this fine-tuning is not analyzed. It is unclear how much fine-tuning is necessary to close the train-inference gap or if more fine-tuning would yield further gains. An ablation on this hyperparameter would strengthen the paper.***
>
> Thank you for this thoughtful question.
> We agree that it is important to analyze how sensitive SPRINT is to the length of the full-token fine-tuning stage and to quantify how much fine-tuning is actually needed to close the train–inference gap. In response, we perform an ablation over the fine-tuning step and added the **qualitative results (Figure 8), quantitative results (Figure 9), and analysis section (Section 4.5)** in the revised paper.
>
> To briefly summarize, we take the model right after 75% drop pre-training, evaluate it with full-token inference (“No Ft”), and then continue training with full tokens for different numbers of iterations. The FID results on ImageNet 256×256 for both CFG and PDG are presented below:
>
> |              | No FT |  20K |  50K | 100K | 150K | 200K |
> |--------------|:-----:|:----:|:----:|:----:|:----:|:----:|
> | SPRINT (CFG) |  3.27 | 1.99 | 1.98 | 1.96 | 1.95 | 1.93 |
> | SPRINT (PDG) |  2.68 | 1.67 | 1.64 | 1.62 | 1.60 | 1.61 |
>
> The main observations are:
> - **The pre-trained model without fine-tuning already has reasonable FID**, confirming that the sparse pre-training stage has learned most of the representation needed for high-quality generation.
> - **Most of the gap is closed very early**—by 20K–50K iterations for both CFG and PDG—when compared to the 200K-iteration endpoint.
> - **Beyond 100K iterations, improvements become marginal and essentially plateau**, with only minor fluctuations across runs.
>
> This shows that SPRINT is not overly sensitive to the exact length of the fine-tuning stage, and that a relatively short full-token fine-tuning (on the order of 20K–50K iterations) is already sufficient to close the train–inference gap. Our default choice of 100K iterations is therefore a conservative setting rather than a strict requirement.
>
> ---
>
> >***Q2. The paper empirically shows that PDG works well by using the shallow path fθ as the unconditional estimate, inspired by Auto Guidance. This is effective, but a deeper conceptual analysis of why fθ is a suitable unconditional predictor would be beneficial. Does the SPRINT training force fθ to learn an ""average"" or ""blurry"" representation that mimics a true unconditional pass? A brief discussion would be insightful.***
>
> Thank you for this insightful question.
> In SPRINT, the shallow path is explicitly optimized as a lower-capacity version of the full model on the *same task and data*, and thus naturally learns a weaker, smoother prediction.
> Concretely, during training we occasionally bypass the entire middle block and update only the **encoder–decoder stack**. This shallow path is optimized to predict the same velocity under the same conditioning as the full model, but with *significantly lower capacity*.
>
> As a result, the shallow path converges to a less strongly conditioned, more “average-like” prediction. This is consistent with our qualitative results in Figure 4: samples generated using only the shallow path are less class-aligned and less semantically structured compared to those from the full model. In this sense, it behaves like the “weaker” denoiser in the AutoGuidance paper, providing a smoother surrogate that can act as an effective unconditional estimate and reduce the errors of the stronger full model.

---

> ### Author Response · Authors · 2025-11-20
>
> >***Q3. The performance of PDG is excellent. Does the SPRINT training enable PDG? In other words, could PDG (i.e., using only the first few layers fθ for the unconditional pass) be applied as an inference-time optimization to a standard, densely trained SiT model, or is the sparse-dense training process necessary for fθ to become an effective unconditional predictor?***
>
> Thank you for this insightful question. PDG is enabled by the dual-path design in SPRINT, but the underlying idea is not restricted to the sparse–dense training setup itself. As mentioned in Q2, this dual-path design allows the shallow-only path to be optimized as a lower-capacity sub-model of the full network during training.
>
> This idea can be transferred to a standard densely trained SiT model, but **not** as a purely inference-time trick. Simply dropping the middle layers of a SiT that was never trained in this dual-path manner yields a sub-model that has not been optimized to produce meaningful predictions. To make PDG work in this setting, we need to add a residual-fusion branch to the SiT architecture and fine-tune it.
>
> In response, we tested this by starting from a pretrained dense SiT-XL model (FID 2.04 with its original CFG), adding a residual fusion branch between the 2nd and 26th layers, and then fine-tuning. The FID of SiT with PDG sampling over fine-tuning steps is:
>
> |       |  FT 0 | FT 100K | FT 200K |
> |-----------|:-----:|:-------:|:-------:|
> | SiT (PDG) | 110.1 |   2.95  |   2.45  |
>
> The results show that SiT with PDG sampling steadily improves as fine-tuning progresses, indicating that PDG can be applied to a standard densely trained SiT model once it is adapted to the dual-path design. However, because SiT was not originally trained with this structure, it requires substantial fine-tuning to make the shallow path an effective unconditional predictor, whereas in SPRINT this behavior emerges naturally from the training procedure.
>
> ---
>
> >***Q4. Could you provide experimental results on text-to-image models (such as PixArt, Flux), similar to the fine-tuning experiments on FLUX, to more convincingly demonstrate the effectiveness of your proposed method?***
>
> Thank you for the suggestion. We agree that demonstrating SPRINT on large-scale text-to-image models such as PixArt or FLUX would be very interesting and would further support the generality of our approach.
>
> In this work, we focused our compute budget on training from scratch on ImageNet with several DiT-style backbones and resolutions. Training text-to-image models such as PixArt or FLUX would require a substantially larger compute budget than we currently have available during the rebuttal period.
>
> That said, SPRINT is architecturally agnostic, and it already works consistently across multiple DiT backbones and resolutions. This suggests that it should transfer naturally to large text-to-image models as well, and we plan to explore this in future extensions.
>
> ---
> We hope our responses clarify the reviewer’s concerns and would be happy to address any further questions.

---

> > ### Comment · Reviewer_VDGb · 2025-11-26
> >
> > Thank you for the clarifications. I am satisfied with the responses and will maintain my score.

---

### Official Review · Reviewer_orVZ · 2025-10-30

**Soundness:** 3
**Presentation:** 3
**Contribution:** 3
**Rating:** 6
**Confidence:** 5

**Summary:**

This paper introduces SPRINT, a novel framework for efficient training of Diffusion Transformers (DiTs) by leveraging sparse-dense residual fusion. It enables aggressive token dropping (up to 75%) while preserving representation quality, significantly reducing training costs (up to 9.8×) and inference FLOPs. SPRINT trains DiTs in two stages: sparse pre-training and short full-token fine-tuning to bridge the train-inference gap. It also introduces Path-Drop Guidance (PDG), a more efficient alternative to classifier-free guidance, further improving generation quality and efficiency. The method is simple, architecture-agnostic, and applicable across various resolutions and models.

**Strengths:**

+ Good performance
+ The proposed Dense shallow path  and sparse deep path can effectively accelerate the training speed.

**Weaknesses:**

1. More discussion on Path-Drop Guidance should be included in the Introduction. Currently, the manuscript treats it as merely a supplementary design.

2. The font size in the tables should be consistent.

**Questions:**

see weakness

---

> ### Author Response · Authors · 2025-11-19
>
> >***Q1. More discussion on Path-Drop Guidance should be included in the Introduction. Currently, the manuscript treats it as merely a supplementary design.***
>
> Thank you for this helpful suggestion. We agree that Path-Drop Guidance (PDG) deserves more prominence in the Introduction, as it is an important practical contribution of SPRINT rather than just a minor supplementary design.
> In the revised manuscript, we’ve updated the introduction section to explicitly highlight PDG as one of the key contributions.
>
> ---
>
> >***Q2. The font size in the tables should be consistent.***
>
> Thank you for pointing this out. We will revise the camera-ready version to ensure that all tables use a consistent font size.
>
> ---
> We hope our responses clarify the reviewer’s concerns and would be happy to address any further questions.

---

> > ### Comment · Reviewer_orVZ · 2025-11-28
> >
> > Thank you for your response. My concerns are addressed.

---

### Official Review · Reviewer_wnyK · 2025-10-31

**Soundness:** 2
**Presentation:** 3
**Contribution:** 2
**Rating:** 2
**Confidence:** 4

**Summary:**

This paper proposes SPRINT, a training method for Diffusion Transformers (DiTs) that aims to reduce training costs through aggressive token dropping (up to 75%). The core idea is to partition the DiT into encoder-middle-decoder components, where the encoder processes all tokens, the middle blocks operate on sparse tokens, and outputs are fused through residual connections. The authors claim 9.8x training savings with comparable quality on ImageNet-1K $256^2$.

**Strengths:**

1. Practical Problem: The paper addresses the important issue of quadratic training costs in DiTs, which is highly relevant for the community.
2. Strong Empirical Results: The reported 9.8x training speedup with maintained quality is impressive if valid.
3. Architecture Agnostic: The method appears to work across different architectures (SiT, UViT) and can be combined with other techniques like REPA.
4. Comprehensive Experiments: The paper includes extensive ablations and analysis across multiple settings.

**Weaknesses:**

Major Concerns

1. Limited Technical Novelty.
The core contribution appears to be a modification of MDTv2, essentially replacing the side-interpolator with simple residual connections. The encoder-middle-decoder architecture is questonable, and the paper fails to provide compelling theoretical or empirical justification for why this specific design should outperform existing methods like MDTv2.

2. Insufficient Comparison with Prior Work.
The paper does not adequately explain why SPRINT should be superior to MDTv2. The fundamental question remains unanswered: what specific advantages does replacing MDTv2's side-interpolator with residual connections provide? The paper lacks rigorous analysis of this key design choice.

3. Questionable Performance Claims.
From Table 3, SPRINT appears to underperform compared to MDTv2 (in terms of FID). This contradicts the paper's claims of superiority and raises questions about the experimental setup and evaluation fairness.

4. Lack of Theoretical Foundation.
The paper provides insufficient theoretical justification for why the proposed encoder-middle-decoder architecture can support such high drop rates (75%). The explanation about shallow vs. deep layer specialization is intuitive but lacks rigorous analysis or proof.

5. Missing Critical Analysis.
The paper doesn't adequately address:
- Why simple residual fusion should be better than more sophisticated fusion mechanisms
- Why SPRINT can tolerate 75% drop rate
- How the method compares to MDTv2 in controlled settings with identical experimental conditions

Minor Issues
- Writing Quality: Some sections lack clarity, particularly the technical description of the fusion mechanism.
- Experimental Setup: More details needed on fair comparison protocols with baseline methods.
- Ablation Studies: While extensive, the ablations don't address the core question of why (if really so) this approach works better than MDTv2.

**Questions:**

1. Can you provide a direct, controlled comparison with MDTv2 using identical experimental settings?
2. What theoretical or empirical evidence supports the advantages of the residual fusion over MDTv2's approach?
3. Why are the FLOPs for MDTv2 missing in Table 3? It has open-sourced official code, you should check it and measure the costs.

---

> ### Author Response · Authors · 2025-11-19
>
> > ***Q1. The core contribution appears to be a modification of MDTv2, essentially replacing the side-interpolator with simple residual connections. The encoder-middle-decoder architecture is questionable, and the paper fails to provide compelling theoretical or empirical justification for why this specific design should outperform existing methods like MDTv2.***
>
> Thank you for the thoughtful question. We would like to clarify that SPRINT’s main contribution is not the *technical choice of a fusion mechanism*, but the underlying **architectural specialization** derived from the complementary roles of shallow and deep layers.
>
> Our motivation is grounded in the observation that **shallow layers** are highly sensitive to noise and are responsible for aggregating local, high-frequency information from **all tokens**.
> In contrast, **deeper layers** primarily learn global, low-frequency semantics that exhibit strong **spatial redundancy**.
> This leads to our core architecture design in which shallow layers remain dense, so that they can transform the noisy input into noise-invariant semantic features, while subsequent middle layers can safely operate on a sparse subset of tokens. The decoder then fuses dense shallow and sparse deep features to produce a dense prediction.
> Concretely, SPRINT follows an **encoder (dense) - middle (sparse) - decoder (dense)** structure.
>
> This is conceptually and architecturally different from MDTv2.
> MDTv2 is designed via masked latent modeling and applies token dropping **directly to the input latent noise**, so its encoder operates on sparse tokens and its decoder on dense tokens, i.e., **encoder (sparse) – decoder (dense)**. To recover information lost at the encoder input, MDTv2 introduces a side-interpolator and additional transformer blocks in the decoder, which increases the total parameter count and training compute (about 10% more parameters than a DiT-XL backbone).
>
> In our main paper. we’ve already demonstrated empirical evidence that our architectural choice is necessary and effective:
> - Figures 3b and 6 show that the encoder in SPRINT effectively learns to transform noisy inputs into noise-invariant representations across diffusion timesteps.
> - Figure 4 qualitatively illustrates that shallow and deep paths capture different but complementary information.
> - Tables 5, 6, and 8 present ablations where we remove or alter the dense-shallow path or the sparse-deep path; in all cases performance degrades, indicating that our proposed encoder-middle-decoder specialization is critical for high drop ratios.
>
> For these reasons, SPRINT should **NOT** be viewed as “MDTv2 with a different fusion operator”. It is based on a distinct architectural principle that enables stable and efficient training at aggressive drop rates beyond what MDTv2 supports.
>
> ---
> > ***Q2. Questionable Performance Claims. From Table 3, SPRINT appears to underperform compared to MDTv2 (in terms of FID). This contradicts the paper's claims of superiority and raises questions about the experimental setup and evaluation fairness.***
>
> We would like to clarify that the MDTv2 numbers in Table 3 do not correspond to an apples-to-apples comparison with SPRINT, and that **SPRINT in fact outperforms MDTv2 under matched settings**.
>
> - **The officially reported MDTv2 performance uses a *stronger sampling strategy***.
> The MDTv2 results employ the timestep-dependent guidance schedule, which is known to boost performance [1]. For a fair comparison with other DiT-based baselines, we initially reported SPRINT with standard fixed scale, so the apparent FID gap comes from different sampling strategies rather than an inferior generative model. As we show in the updated **Table 3 in above “Global Response”**, when we evaluate SPRINT under the guidance schedule, it achieves better FID and FDD than MDTv2.
>
> - **The official MDTv2 checkpoint is trained with a *significantly larger compute*** (1080 epochs vs. our 200–400). Even under this compute advantage, our model achieves a stronger efficiency–quality trade-off, as reflected in FDD (58.4 ours VS. 75.2 MDTv2) and FLOPs.
>
> - To remove all confounding factors, we re-implemented MDTv2 using its official code and trained it under identical settings to SPRINT. As reported in the updated **Table 1 in above “Global response”**, under this controlled setup SPRINT achieves substantially better performance in both FID and FDD than MDTv2.
>
> ---
> [1] Karras, Tero, et al, "Guiding a Diffusion Model with a Bad Version of Itself", NeurIPS 2024

---

> > ### Author Response · Authors · 2025-11-19
> >
> > >***Q3. Lack of Theoretical Foundation. The paper provides insufficient theoretical justification for why the proposed encoder-middle-decoder architecture can support such high drop rates (75%). The explanation about shallow vs. deep layer specialization is intuitive but lacks rigorous analysis or proof.***
> >
> > Thank you for this comment. We would like to note that the goal of SPRINT is to propose and validate an empirically effective architectural design for efficient training, rather than to provide a formal theory of token dropping optimality. We note that existing token-dropping and masked diffusion models—including MDTv2, MaskDiT, MicroDiT, and TREAD—also do not provide theoretical guarantees for their masking strategies; prior work in this area is largely empirical.
> >
> > Token dropping in diffusion transformers is closely related to token pruning in vision transformers [2,3,4] or large language models [5, 6]. Across these domains, the standard practice is to justify designs through empirical analysis and ablations, not through formal proofs for specific drop ratios. In this sense, SPRINT follows the prevailing methodology in the field.
> >
> > Within this empirical framework, we provide several forms of evidence supporting our encoder–middle–decoder architecture at high drop rates:
> > - **Sections 3.2 and 3.3** present a clear motivation based on shallow–deep specialization: shallow layers are kept dense to preserve local, noise-sensitive information, while deeper layers, which model more redundant global semantics, are made sparse.
> > - We include analyses such as gradient-norm behavior and internal representation similarity across different drop ratios (**Fig. 3a–b**), which show that higher drop ratios emphasize learning in the dense shallow path while maintaining stable deep representations.
> > - We conduct thorough ablation studies (**Tables 5–8**) that systematically vary the architecture and drop configuration and show that the proposed encoder–middle–decoder structure is necessary for stable and high-quality generation at 75% drop.
> >
> > ---
> > >***Q4. Why simple residual fusion should be better than more sophisticated fusion mechanisms.***
> >
> > Thank you for the question. We would like to clarify that our paper does not claim that simple residual fusion is universally superior to all more sophisticated fusion mechanisms. Rather, our goal is to show that, under the proposed architectural specialization, a lightweight fusion module is sufficient to achieve strong performance and stable training at high drop rates, while keeping both parameter count and compute overhead minimal.
> >
> > Exploring more elaborate fusion schemes on top of SPRINT is an interesting avenue for future work, but it is orthogonal to the main contribution of the paper.
> >
> > ---
> > >***Q5. Why SPRINT can tolerate 75% drop rate?***
> >
> > The ability of SPRINT to tolerate a 75% drop rate is a direct consequence of how we distribute computation across network depth.
> > In SPRINT, the shallow layers are kept fully dense and are responsible for aggregating local, noise-sensitive information into noise-invariant features, so that subsequent middle layers can safely operate on a sparse subset of tokens without losing critical detail.
> > The decoder always receives both the dense shallow representation and the sparse deep representation, so dropped tokens affect only the global semantic refinement, not the reconstruction of local structure.
> >
> > Empirically, as shown in our ablations (in Table 5-8), alternative architectures either collapse or degrade severely at 75% drop, whereas SPRINT remains stable and continues to improve, supporting this design choice.
> >
> > ---
> > >***Q6. How the method compares to MDTv2 in controlled settings with identical experimental conditions.***
> >
> > Please refer to the updated Table 1 in “Global Response”.
> >
> > ---
> > >***Q7. Why are the FLOPs for MDTv2 missing in Table 3? It has open-sourced official code, you should check it and measure the costs.***
> >
> > Thank you for pointing this out. In response to your suggestion, we have updated both training and inference FLOPs for MDTv2 to Table 3 in the revised manuscript, using the official MDTv2 checkpoint.
> >
> > Concretely, MDTv2-XL requires 258.3 TFLOPs for training, which is **roughly 4× larger than** our SPRINT-XL (65.1 TFLOPs). For inference, MDTv2-XL requires 0.521 TFLOPs per sample, which is about **2× larger than** our PDG sampling.
> >
> >
> > ---
> > [2] Rao, Yongming, et al. "Dynamicvit: Efficient vision transformers with dynamic token sparsification.", NeurIPS 2021
> > [3] Bolya, Daniel, et al. "Token merging: Your vit but faster.", ICLR 2023
> > [4] Chang, Shuning, et al. "Making vision transformers efficient from a token sparsification view.", CVPR 2023
> > [5] Hou, Le, et al. "Token dropping for efficient bert pretraining." arXiv 2022
> > [6] Nawrot, Piotr, et al. "Efficient transformers with dynamic token pooling." ACL 2023

---

> ### Comment · Reviewer_wnyK · 2025-11-26
>
> The authors have addressed most of my concerns. However, I suggest that the authors explain in the manuscript how the specific architecture design enables a high token drop ratio, rather than just describing the phenomenon.

---

### Official Review · Reviewer_pKvE · 2025-11-01

**Soundness:** 3
**Presentation:** 3
**Contribution:** 3
**Rating:** 6
**Confidence:** 2

**Summary:**

This paper proposes SPRINT, a method to accelerate Diffusion Transformer (DiT) training through sparse-dense residual fusion. SPRINT processes all tokens in shallow layers for local details, drops 75% of tokens in deep layers for global semantics, and fuses outputs via residual connections. Training uses sparse pre-training followed by brief full-token fine-tuning. On ImageNet-1K 256×256, SPRINT achieves 9.8× training speedup with comparable quality.

**Strengths:**

(1)SPRINT achieves 9.8× training speedup on ImageNet-1K while maintaining comparable quality. The method adds only 0.3% parameters and preserves standard DiT blocks, making it easy to integrate. Strong generalization across architectures (SiT, U-ViT, REPA) demonstrates practical value.
(2)Path-Drop Guidance (PDG) halves inference FLOPs while improving quality. Comprehensive experiments reveal complementary roles of sparse-deep and dense-shallow features, providing valuable insights into DiT representation mechanisms and explaining why the sparse-dense fusion design is effective.

**Weaknesses:**

(1)The paper claims that two-stage training can "close the train-inference gap," but does not quantify how large this gap actually is.

**Questions:**

(1)Why is there no comparative validation using features from different layers for unconditional guidance?
(2)After pre-training with 75% drop ratio, if full-token inference is performed directly (without fine-tuning), how much would the performance degrade?

---

> ### Author Response · Authors · 2025-11-19
>
> > ***Q1. The paper claims that two-stage training can "close the train-inference gap," but does not quantify how large this gap actually is. After pre-training with 75% drop ratio, if full-token inference is performed directly (without fine-tuning), how much would the performance degrade?***
>
> Thank you for this thoughtful question. We agree that it is important to quantify the gap between the token-dropping pre-training stage and the final full-token model. In response, we perform an ablation over the fine-tuning step and added the **qualitative results (Figure 8), quantitative results (Figure 9), and analysis section (Section 4.5)** in the revised paper.
>
> To briefly summarize, we take the model right after 75% drop pre-training, evaluate it with full-token inference (“No Ft”), and then continue training with full tokens for different numbers of iterations. The FID results on ImageNet 256×256 for both CFG and PDG are presented below:
>
> |              | No FT |  20K |  50K | 100K | 150K | 200K |
> |--------------|:-----:|:----:|:----:|:----:|:----:|:----:|
> | SPRINT (CFG) |  3.27 | 1.99 | 1.98 | 1.96 | 1.95 | 1.93 |
> | SPRINT (PDG) |  2.68 | 1.67 | 1.64 | 1.62 | 1.60 | 1.61 |
>
> The main observations are:
> - **The pre-trained model without fine-tuning already has reasonable FID**, confirming that the sparse pre-training stage has learned most of the representation needed for high-quality generation.
> - **Most of the gap is closed very early**—by 20K–50K iterations for both CFG and PDG—when compared to the 200K-iteration endpoint.
> - **Beyond 100K iterations, improvements become marginal and essentially plateau**, with only minor fluctuations across runs.
>
> This shows that SPRINT is not overly sensitive to the exact length of the fine-tuning stage, and that a relatively short full-token fine-tuning (on the order of 20K–50K iterations) is already sufficient to close the train–inference gap.
>
> ---
>
> > ***Q2. Why is there no comparative validation using features from different layers for unconditional guidance?***
>
> We understand this question as referring to using features from different depths of the middle block to compute the unconditional velocity in PDG sampling, instead of completely bypassing the middle block for the unconditional branch.
>
> In response, we have tested using an intermediate feature of the first few middle blocks to compute the unconditional velocity.
> Empirically, we observe that using intermediate features significantly degrades the performance. This is because the model, especially the decoder, is not trained to estimate accurate velocity given intermediate features.
> Therefore, when we feed intermediate features into the decoder, the resulting unconditional velocity severely deviates from the learned manifold, leading to unexpected results.
>
> ---
>
> We hope our responses clarify the reviewer’s concerns and would be happy to address any further questions.

---

> > ### Comment · Reviewer_pKvE · 2025-11-26
> >
> > Thank you for the clarifications. I will maintain my score.

---

> > > ### Author Response · Authors · 2025-11-27
> > >
> > > Thank you again for your response and for taking the time to read our clarifications.
> > >
> > > We hope we have addressed your main concerns through (1) the updated results quantifying the train–inference gap, and (2) the additional validation using features from different layers for unconditional guidance.
> > >
> > > Given these new results and revisions, we would appreciate it if you could consider whether your current score still reflects your updated assessment of the paper. If there are any remaining questions or further feedback that can improve the paper, we would be very happy to discuss them. Thank you again for your careful review and engagement during the rebuttal.

---

### Author Response · Authors · 2025-11-19
**Global Response**

We thank all reviewers for their time and effort in evaluating our paper.
In particular, we would like to inform you that we have ***updated Tables 1 and 3*** in the main paper in response to Reviewer wnyK’s concern regarding the comparison with MDTv2.
These updates provide a more comprehensive and controlled comparison between our SPRINT and MDTv2.
We briefly summarize the changes below; please refer to the Table 1 and Table 3 in revised manuscript for the complete results.

---

### **< Table 1 >**
- We ran MDTv2 with the same architectural improvements and flow-matching objective as SPRINT, increasing its token drop ratio from the original 0.35–0.5 to 0.75.
- The updated Table 1 shows that SPRINT ***consistently outperforms*** MDTv2 across all metrics, despite using ***less training compute***, demonstrating the effectiveness of our method.

|               |                |           |  w/o CFG |           |   |           |   w CFG  |           |
|:-------------:|:--------------:|:---------:|:--------:|:---------:|:-:|:---------:|:--------:|:---------:|
|     Method    | TFLOPS (x10^6) |    FDD    |    FID   |     IS    |   |    FDD    |    FID   |     IS    |
|     MDTv2     |      21.1      |   558.5   |   21.1   |    68.9   |   |   366.5   |   5.61   |   176.3   |
| **SPRINT (Ours)** |    **18.7**    | **262.6** | **9.30** | **118.5** |   | **136.5** | **2.56** | **247.1** |


---

### **< Table 2 >**
- We add evaluation results of the official MDTv2 checkpoint with vanilla CFG sampling (without the guidance schedule).
Under this setting, SPRINT-PDG outperforms MDTv2 by a large margin in both FDD (58.4 vs. 77.3) and FID (1.62 vs. 1.86), while using substantially less training compute (65.1 vs. 258.3 TFLOPs).

- We add evaluation results of SPRINT-PDG with guidance schedule strategy. With guidance schedule, SPRINT-PDG achieves better FDD (54.9 vs. 75.2) and FID (1.55 vs. 1.58) than MDTv2.

- We also report SPRINT$_{\text{PDG+REPA}}$ with guidance schedule, which achieves **FDD = 49.6** and **FID = 1.49** with only 400 training epochs, further highlighting the strong efficiency–quality trade-off of our approach.


|         Method        |  Epochs |  #Params | Train TFLOPS ($\times10^6$) | Inference TFLOPs |    FDD $\downarrow$   |    FID $\downarrow$  |
|---------------------|:-------:|:--------:|:--------------------:|:----------------:|:--------:|:--------:|
| MDTv2-XL              |   1080  |   742M   |         258.3        |       0.521      |   77.3   |   1.86   |
| MDTv2-XL$^{\ddagger}$          |   1080  |   742M   |         258.3        |       0.521      |   75.2   |   1.58   |
|   SPRINT$_{\text{PDG}}$ **(Ours)**   | **400** | **677M** |       **65.1**       |     **0.274**    | **58.4** | **1.62** |
| SPRINT$_{\text{PDG}}^{\ddagger}$ **(Ours)** | **400** | **677M** |       **65.1**       |     **0.263**    | **54.9** | **1.55** |
| SPRINT$_{\text{PDG+REPA}}^{\ddagger}$ **(Ours)** | **400** | **677M** |       **66.7**       |     **0.263**    | **49.6** | **1.49** |

---

### Public Comment · ~Felix_Krause1 · 2025-11-25
**Comparison to our method**

Thanks for making this work available.

One small note: some of the baseline comparisons including the one involving our method, TREAD (Krause et al., 2025), appear to use evaluation settings that differ from those used in the original paper. In particular, TREAD is evaluated here with a 75% selection / masking rate, a configuration that we explicitly reported as degrading performance. In all our main experiments, we instead use a 50% selection rate as the default. Under this configuration (75%), the numbers shown for TREAD in Table 1 (and the teaser figure) do not reflect the intended or recommended usage of the method, as we reach an FID of **4.9** under our default setting (50%) versus the **16.3** reported here for the 75% configuration.

If these modifications are necessary for your setup, it might be helpful for future readers to make them more explicit in the main text or in the figure/table captions, and to clearly distinguish between “default” and “stress-test” regimes, especially since convergence speed and training efficiency are central claims of the paper. In the same spirit, we would kindly recommend also including results for TREAD under its default 50% selection rate, rather than only under a modified setting that is known to yield weaker performance, so that readers can more easily assess both convergence speed and final quality under the intended configuration of the method.

Sharing this observation publicly in case it’s helpful for readers trying to understand how these methods compare.

---

> ### Author Response · Authors · 2025-11-27
>
> Thank you for pointing this out. We would like to clarify that we did not intend to weaken the TREAD baseline.
>
> In Table 1, all token-dropping methods (including TREAD) are evaluated with a 75% drop ratio. This was a deliberate design choice to study the aggressive token-dropping regime that motivates SPRINT, i.e., stable and efficient training under very high drop ratios. We apply the same 75% setting to all token-dropping baselines for a controlled comparison, and this is explicitly stated in the caption of Table 1 of our revision; we will further sharpen the wording to make this clear as a “75% stress-test” setting.
>
> At the same time, we also report each method under its recommended configuration in Table 3, which is intended to show best-achievable quality. In particular, we include the officially reported TREAD result at its default 50% selection rate, so that readers can see its performance under the intended setting.
>
> For the teaser in Figure 1, we could not report the official performance of TREAD because no official checkpoint is publicly available. In response to your suggestion, we will train TREAD with the default 50% drop ratio and add these numbers to Figure 1.

---

### Author Response · Authors · 2025-12-02
**Rebuttal Summary for Area Chairs - Part 2/2**

> **3. Summary of new experiments during rebuttal**

During the rebuttal period, we invested substantial effort into running new experiments and adding analyses requested by the reviewers.
We summarize the main additions below:

| New experiments                                                                                 | Key points                                                                                                                                                                                                                                                  | Concerns addressed                                                                                 |
|-------------------------------------------------------------------------------------------------|-------------------------------------------------------------------------------------------------------------------------------------------------------------------------------------------------------------------------------------------------------------|----------------------------------------------------------------------------------------------------|
| Train MDTv2 under identical settings as SPRINT (in **Table 1**)                                     | SPRINT consistently outperforms MDTv2 across all metrics in identical setting, despite using less training compute.                                                                                                                                         | **wnyK**: novelty and fair/complete comparison to MDTv2                                                |
| Evaluate SPRINT with guidance schedule sampling (in **Table 3**)                                    | SPRINT achieves better FDD (54.9 vs. 75.2) and FID (1.55 vs. 1.58) than MDTv2.                                                                                                                                                                              | **wnyK**: fair/complete comparison to MDTv2                                                            |
| Evaluate training-inference gap: Qualitative, quantitative, additional section (in **Section 4.5**) | We add a dedicated subsection and new figures/tables analyzing the fine-tuning stage.  Results show that SPRINT is not overly sensitive to the fine-tuning length: a relatively short full-token fine-tuning already closes most of the train–inference gap | **pKvE, VDGb**: train–inference gap magnitude & sensitivity to fine-tuning length                      |
| Train PDG to standard SiT                                                                       | Demonstrates PDG can be applied to a standard densely trained SiT model once adapted to a dual-path design.                                                                                                                                                 | **VDGb**: generality of PDG to standard SiT                                                            |
| Train TREAD at its recommended 50% drop ratio (**Figure 1**)                                        | We additionally evaluate TREAD under its default 50% drop rate. This provides a fair view of TREAD under its intended setting and does not change our core conclusion that SPRINT offers a stronger efficiency–quality trade-off.                           | **Public comment**: fairness of baseline configuration                                                 |
| Re-emphasize our architectural analysis (in **Figs. 3b, 4, 6** and **Tabs. 5-8**)                       | We highlight that we provide extensive analysis of SPRINT and the roles of dense–shallow and sparse–deep paths, directly supporting our architectural motivation, novelty claims, and ablations                                                            | **wnyK**: Novelty, justification of SPRINT’s architecture and effectiveness, missing critical analysis |

---

We hope this summary clarifies our contributions and helps your final decision. We sincerely thank you again for your time and careful evaluation.

---

### Author Response · Authors · 2025-12-02
**Rebuttal Summary for Area Chairs - Part 1/2**

We deeply appreciate the ACs for carrying an exceptionally heavy workload and pressure given the unusual reviewing circumstances this year. We are sincerely grateful for your time and careful evaluation. To make your decision easier, we would like to briefly summarize our paper and rebuttal:
- A short summary of our paper’s motivation and contributions
- A summary of reviewers’ key concerns
- A summary of our efforts and new experiments addressing those concerns
---

> **1. Our paper’s motivation and novelties**

Our paper, **SPRINT**, tackles the severe computational bottleneck of training diffusion transformers by aggressively dropping tokens during training, while preserving generation quality. SPRINT achieves up to **9.8× savings in training compute** while maintaining or surpassing the performance of strong DiT baselines.

Concretely, our main novelties are:
- **Sparse–dense residual fusion architecture**:
We introduce an architectural computation specialization that explicitly assigns shallow layers to dense tokens and deep layers to sparse tokens and merges both paths with residual fusion. This design enables ***stable and effective training at 75% token drop***, where prior token-dropping and masked-latent methods ***either fail or degrade severely***.

- **Path Drop guidance (PDG)**:
Leveraging the dual-path structure, we propose PDG, which uses a shallow-only path as a cheap surrogate for the unconditional branch. PDG nearly ***halves the inference cost*** of classifier-free guidance while improving FID/FDD compared to standard CFG, without training a separate unconditional model.

In the main paper, we detail the motivation and rationale behind our design (**Section 3.2 and 3.3**), supported by comprehensive empirical evidence. This demonstrates that our architectural choice is necessary and effective for aggressive token-dropping:
- **Figures 3b** and **6** show that the encoder in SPRINT effectively learns to transform noisy inputs into noise-invariant representations.
- **Figure 4** qualitatively illustrates that shallow and deep paths capture different but complementary information.
- **Tables 5, 6**, and **8** present ablations where we remove or alter the dense-shallow path or the sparse-deep path; in all cases performance degrades, indicating that our proposed design is critical for high drop ratios.

---

> **2. Summary of reviewer’s attitude and concern**

We appreciate the thoughtful comments from all four reviewers, which have helped us meaningfully refine and strengthen the paper.
Overall, the initial reception was **positive**, e.g., *“significant training efficiency in a practical problem”*, *“strong results,”* *“simplicity and generality,”* *“novel and effective inference acceleration (PDG),”* and *“comprehensive experiments.”*

We summarize the key concerns raised by the reviewers in the table below:

| Reviewer | Rating | Key concern before rebuttal                                                                                                                      | After rebuttal                                                                    |
|----------|:------:|--------------------------------------------------------------------------------------------------------------------------------------------------|-----------------------------------------------------------------------------------|
| **pKvE**     |    6   | Train–inference gap (its size and required fine-tuning); additional validation of PDG (features from different layers).                          | The reviewer notes: “*Thank you for the clarifications. I will maintain my score.*” |
| **wnyK**     |    2   | Novelty and fair/complete comparison to MDTv2; justification of SPRINT’s architecture and effectiveness; missing critical analysis               | The reviewer notes: “*The authors have addressed most of my concerns.*”             |
| **orVZ**     |    6   | Emphasis on PDG in the Introduction; table font-size consistency.                                                                                | The reviewer notes: “*My concerns are addressed.*”                                  |
| **VDGb**     |    8   | Sensitivity to fine-tuning length / train–inference gap; generality of PDG to standard SiT; large-scale text-to-image experiments (PixArt/FLUX). | The reviewer notes: “*I am satisfied with the responses.*”                           |

---

### Meta-Review · Area_Chair_aiR8 · 2026-01-07

**Summary:**

The reviews are positive overall: three reviewers (scores 8, 8, 6) highlight the strong empirical results, significant practical efficiency gains (9.8× training speedup with comparable or better quality), simplicity and generality of the sparse-dense fusion design, novel inference acceleration via Path-Drop Guidance (PDG), and comprehensive experiments/ablation studies. One reviewer (initial score 2) raised concerns about novelty and baseline fairness. The authors' rebuttal provided substantial new experiments (controlled MDTv2 training under identical settings, detailed train-inference gap analysis with new figures/tables/subsection, PDG application to standard SiT, TREAD at default 50% drop, and guidance schedule evaluations) and clarifications, convincingly addressing nearly all concerns and strengthening the paper.

The authors are expected to incorporate all new results and feedbacks during the reviewing process into either the main paper or the appendix.

**Reviewer Concerns:**

All major concerns were effectively addressed in the rebuttal:

- Fair and complete comparison to MDTv2, including novelty (new controlled experiments showing SPRINT superiority despite lower compute);
- Train-inference gap magnitude, sensitivity, and fine-tuning requirements (new subsection, figures, and tables demonstrating short fine-tuning suffices);
- PDG generality and validation (applied to standard SiT; intermediate feature ablations);
- Baseline fairness for TREAD (added default 50% results);
- Emphasis on PDG, architectural justification, and minor presentation issues (clarified and updated).

Still outstanding: None major; the initially skeptical reviewer acknowledged that most concerns were addressed.

**Reviewer Scores:**

- Reviewer pKvE (initial 6): Likely unchanged (stated "I will maintain my score").
- Reviewer wnyK (initial 2): Likely increasing (stated "authors have addressed most of my concerns").
- Reviewer orvZ (initial 6): Likely unchanged or slight increase (concerns fully addressed).
- Reviewer VDgb (initial 8): Likely unchanged (fully satisfied).

---

### Decision · Program_Chairs · 2026-01-26

Accept (Poster)